# Steel, Aluminum, and FRP-Composites: The Race to Zero Carbon Emissions

Vaishnavi Vijay Rajulwar [1], Tetiana Shyrokykh [1,*], Robert Stirling [1], Tova Jarnerud [1,2], Yuri Korobeinikov [1], Sudip Bose [3], Basudev Bhattacharya [4], Debashish Bhattacharjee [3] and Seetharaman Sridhar [1]

[1] Ira A. Fulton Schools of Engineering, Arizona State University, Tempe, AZ 85281, USA; vrajulwa@asu.edu (V.V.R.); robert.stirling@asu.edu (R.S.); tova.jarnerud@swerim.se (T.J.); ikorobei@asu.edu (Y.K.); seetharaman@asu.edu (S.S.)
[2] Swedish Research Institute for Mining, Metallurgy and Materials, Process Metallurgy, SE-974 37 Luleå, Sweden
[3] Tata Steel Ltd., Chowringhee 700071, West Bengal, India; sudip.bose@tatasteel.com (S.B.); debashish.bhattacharjee@tatasteel.com (D.B.)
[4] Tata Steel Ltd., Jamshedpur 831001, Jharkhand, India; basudev@tatasteel.com (B.B.)
* Correspondence: tetiana.shyrokykh@asu.edu

**Abstract:** As various regions around the world implement carbon taxes, we assert that the competitiveness of steel products in the marketplace will shift according to individual manufacturers' ability to reduce $CO_2$ emissions as measured by cradle-to-gate Life Cycle Analysis (LCA). This study was performed by using LCA and cost estimate research to compare the $CO_2$ emissions and the additional cost applied to the production of various decarbonized materials used in sheet for automotive industry applications using the bending stiffness-based weight reduction factor. The pre-pandemic year 2019 was used as a baseline for cost estimates. This paper discusses the future cost scenarios based on carbon taxes and hydrogen cost. The pathways to decarbonize steel and alternative materials such as aluminum and reinforced polymer composites were evaluated. Normalized global warming potential (nGWP) estimates were calculated assuming inputs from the current USA electricity grid, and a hypothetical renewables-based grid. For a current electricity grid mix in the US (with 61% fossil fuels, 19% nuclear, 20% renewables), the lowest nGWP was found to be secondary aluminum and 100% recycled scrap melting of steel. This is followed by the natural gas Direct Reduced Iron–Electric Arc Furnace (DRI-EAF) route with carbon capture and the Blast Furnace-Basic Oxygen Furnace (BF-BOF) route with carbon capture. From the cost point of view, the current cheapest decarbonized production route is natural gas DRI-EAF with Carbon Capture and Storage (CCS). For a renewable electricity grid (50% solar photovoltaic and 50% wind), the lowest GWP was found to be 100% recycled scrap melting of steel and secondary aluminum. This is followed by the hydrogen-based DRI-EAF route and natural gas DRI-EAF with carbon capture. The results indicate that, when applying technologies available today, decarbonized steel will remain competitive, at least in the context of automotive sheet selection compared to aluminum and composites.

**Keywords:** materials cost; life cycle analysis; decarbonization; steel manufacturing; aluminum manufacturing; carbon fiber composite manufacturing; glass fiber composite manufacturing

## 1. Introduction

In October 2014, the European Council established a policy framework for climate and energy, targeting a reduction in greenhouse gas emissions by 40% by 2030 and at least 80% by 2050 compared to 1990 levels, with the objective of preventing the rise in global temperatures above 2 °C [1,2]. Iron and steel manufacturing are among the world's top industrial carbon dioxide emitters, accounting for 25–30% of the manufacturing sector and 6–8% of overall worldwide emissions [3,4]. As various regions around the world implement carbon taxes, it is likely that the competitiveness of steel products in the marketplace will

shift according to individual manufacturers' ability to reduce carbon dioxide emissions as measured by Life Cycle Analysis (LCA). The carbon emissions and decarbonization efforts for substitute materials such as aluminum, glass, plastics, carbon fiber composites, and engineered wood products add to the situation's complexity. In some instances, a steel product may have no practical non-steel competitors, and only regulatory effects will be felt. In others, designers may substitute "green" steel or these other materials for traditionally manufactured steel to appeal to customer preferences or to avoid carbon taxes. There are also different paths for achieving decarbonized steel and their economic competitiveness with one another and the substitute materials will shift depending on the availability of green affordable power at scale, hydrogen gas ($H_2$) production at scale, and cost of Carbon Capture, Usage, and Storage (CCUS).

This study was performed using Life Cycle Analysis (LCA), manufacturing cost estimates, assumptions about carbon taxes, and material properties to compare the cost competitiveness of several materials when manufactured for use in sheet applications. LCA is a robust method to evaluate the multiple environmental impacts over the life stages of a material, product, or process. This effort uses LCA to estimate the Global Warming Potential (GWP) of a variety of materials to estimate the carbon taxes, which would likely be applied in a Carbon Tax or Carbon Cap and Trade Scenario. In the current work, the authors consider steel obtained via different traditional and alternative production routes, as well as aluminum, glass- and carbon-fiber-reinforced composites in the context of their application as automotive body materials.

LCAs for the traditional steelmaking processes have already been broadly studied by different scientists. Neugebauer et al. [5], Burchart-Korol [6], and Backes et al. [7] studied LCAs for conventional Blast Furnace–Basic Oxygen Furnace (BF-BOF) steel production route. The published GWP values range from 1.7 t $CO_{2\,eq}$/t steel to 2.5 t $CO_{2\,eq}$/t steel. In contrast, the production of steel via the scrap-based Electric Arc Furnace (EAF) route gives considerably less $CO_2$ emissions. The GWP of EAF is critically dependent on two parameters: (1) the share of the ore-based materials in the melt charge as they are used for scrap dilution, and (2) power grid mix, as it is the main source of energy for melting. Therefore, it is reasonable to compare the studies which analyzed 100% or near that scrap charge into EAF. The $CO_2$ emissions from the manufacturing of steel using scrap are estimated to have an intensity of 0.3 t $CO_{2\,eq}$/t crude steel by the International Energy Agency (IEA) [8]. When the source material is 100% scrap, steel production in the EAF emits 0.35 t $CO_{2\,eq}$/t of steel according to Kirschen et al. [9], and Birat et al. reported 0.36 t $CO_{2\,eq}$/t steel [10]; whereas, [11] reported 0.44 t $CO_{2\,eq}$/t steel and Kopfle et al. [12] reported 0.47 t $CO_{2\,eq}$/t crude steel, respectively.

An alternative ironmaking technology, namely Direct Reduction of Iron (DRI), is also widely studied. Despite the variety of alternatives to the BF-BOF route, the current study considers only shaft reactors with natural gas (NG) or hydrogen as a reduction agent. The most widespread shaft reactor technology to date is MIDREX, which operates on natural gas or natural gas enriched with $H_2$. This technology is taken as a reference. NG-DRI followed by EAF typically produces 0.82 to 1.16 t $CO_{2\,eq}$/t DRI [13]. The same findings are reported in several GWP investigations of NG-DRI. According to Barati [14], the emissions can range between 1.3 and 1.5 t $CO_{2\,eq}$/t DRI, but Ameling et al. report 1.3 t $CO_{2\,eq}$/t DRI [15]. Even though there are at least two projects exploring in pilot scale hydrogen-based DRI production (HYBRIT in Sweden [16,17] and GISH in the USA [18]), hydrogen-based DRI production is not commercially operated yet. In August 2021, SSAB steel manufacturer produced the first fossil-free steel in the world using HYBRIT technology. Some prior industrial experience with $H_2$-based fluidized bed reactor technology, Circored in Trinidad, exists [19]. Therefore, there are only estimates on how much GWP will be associated with $H_2$-DRI based steelmaking. There have been several studies performed to determine the total $CO_2$ emissions from $H_2$-DRI. The $CO_2$ emissions can range from 0.1 t $CO_{2\,eq}$/t DRI when the GWP from electricity is 0.01 kg $CO_{2\,eq}$/kWh to 1.1 t $CO_{2\,eq}$/t DRI when the GWP from electricity is 0.3 kg $CO_{2\,eq}$/kWh, according to the research of Rechberger et al. [20].

Using hydrogen produced by renewable power releases 0.38 t $CO_{2\,eq}$/t DRI, according to Fan and Friedmann [21]. Direct electrolysis of iron from ores is not considered as an alternative since the authors believe that the technology, while promising, still needs to be demonstrated for industrial scalability.

Aluminum production is much more energy-intensive than steelmaking. The average $CO_2$ emissions from primary aluminum production in North America in 2019 were 8.2 t $CO_{2\,eq}$/t Al and required 38.19 kWh of energy, as reported by the Aluminum Association [22]. Using 2.5 kWh/t of energy, secondary aluminum produced 0.55 t $CO_{2\,eq}$/t Al [22]. According to a different study by Stolz and Frischknecht [23], the manufacturing of primary aluminum produces 9.31 t $CO_{2\,eq}$/t Al, while the production of secondary aluminum from recycled materials produces 0.85 t $CO_{2\,eq}$/t Al. The International Aluminium Institute [24] estimated that in 2020, primary aluminum production would produce 11.2 t $CO_2$ eq/t Al and secondary aluminum production would produce 0.2 kg $CO_{2\,eq}$/t Al. It should be noted that the GWP potential of Al production is highly dependent on the local electricity source, since most of the $CO_2$ footprint indirectly belongs to the electrolysis as the most energy-intensive production step. Usually, aluminum plants are located near cheap hydropower, which is almost carbon-free. But, in recent decades, most of the world's Al capacities were predominantly added in China where cheap coal-based generation is used, resulting in growth of $CO_2$ emissions of the average aluminum product [25]. However, the least GWP claimed for virgin aluminum smelting based on hydropower is in the range of 2.0–4.0 t $CO_{2\,eq}$/t Al [26,27].

The global warming potential of carbon fibers has been studied previously in the work of Kawajiri and Sakamoto and is reported to decline as production scale increases; it emits 43.32 t $CO_{2\,eq}$/t carbon fiber at a 500 ton per year production scale and 24.83 t $CO_{2\,eq}$/t carbon fiber at a 3000 ton per year production scale [28]. Glass fiber manufacturing produces 1.8–4.6 t $CO_{2\,eq}$/t glass fiber and 4.9 t $CO_{2\,eq}$/t resin, as reported by Song et al. [29] and Schiller in Tchana et al. [30].

To reduce carbon dioxide emissions from traditional manufacturing processes, Carbon Capture and Storage (CCS) is studied. Many heavy industries such as chemical and refinery plants use fossil fuel as an energy source. As a result, the manufacturing sector may exhibit greater reluctance toward the switch to a low-emission process than the electricity sector [31]. This study assesses the impact of CCS on the production cost of steel, aluminum, glass-fiber-reinforced composites (GFRC), and carbon-fiber-reinforced composites (CFRC) in reducing carbon dioxide emissions using traditional processes. With an emphasis on the price of carbon capture, this study gives a thorough techno-economic analysis (TEA) of mono-ethanolamine (MEA)-based post-combustion capture methods. Furthermore, the impact of market and regulatory changes on steel's relative competitiveness against the replacements mentioned and the pathways to decarbonize steel, aluminum, and reinforced polymer composites, as well as the methods of their production and environmental impact, were evaluated for various energy sources.

## 2. Methods

### 2.1. Electricity Grid Mix and Energy

To study the global warming potential caused by the production of various materials, different production processes using different energy sources were investigated. In this study, natural gas was considered as the energy source for process heat. According to the U.S. Energy Information Administration, the conventional electricity grid in the USA consists of 61% fossil fuels (natural gas, coal, petroleum), 19% nuclear, and 20% renewables [32]. The hypothetical renewables-based electricity grid consists of 50% solar photovoltaic and 50% wind-generated electricity [33]. To obtain the final $CO_2$ emissions per kg of material, the $CO_2$ embedded in the energy grid is added to direct emissions and other process emissions. For each manufacturing process and material, the GWP (Global Warming Potential) is calculated using coefficients of 0.385 kg $CO_{2\,eq}$/$kWh_e$ [34] for the traditional grid and 0.032 kg $CO_{2\,eq}$/$kWh_e$ [33] for the renewables-based grid. We assume

a price of USD 0.0545/kWh for both the traditional and renewables-based grids, which corresponds to the 2019 average industrial rate in Texas, USA. Texas was chosen as the analysis region given its intensive industrial development and high potential for carbon capture and storage projects.

### 2.2. Life Cycle Analysis of Steel Production Processes

### 2.2.1. Blast Furnace and Basic Oxygen Furnace

The BF-BOF process is a primary method used in the iron and steel industry to produce steel by reducing iron ore and smelting it into pig iron in a blast furnace and then converting pig iron into steel in a basic oxygen furnace. This process relies heavily on coal products and results in significant $CO_2$ emissions. The BF-BOF process is the main contributor to world crude steel production, accounting for 70.8% of global production [35]. Integrated BF-BOF operations include such processes as coke making, pelleting, or sintering, iron making and steelmaking, continuous casting, and hot rolling into steel sheet (Figure 1).

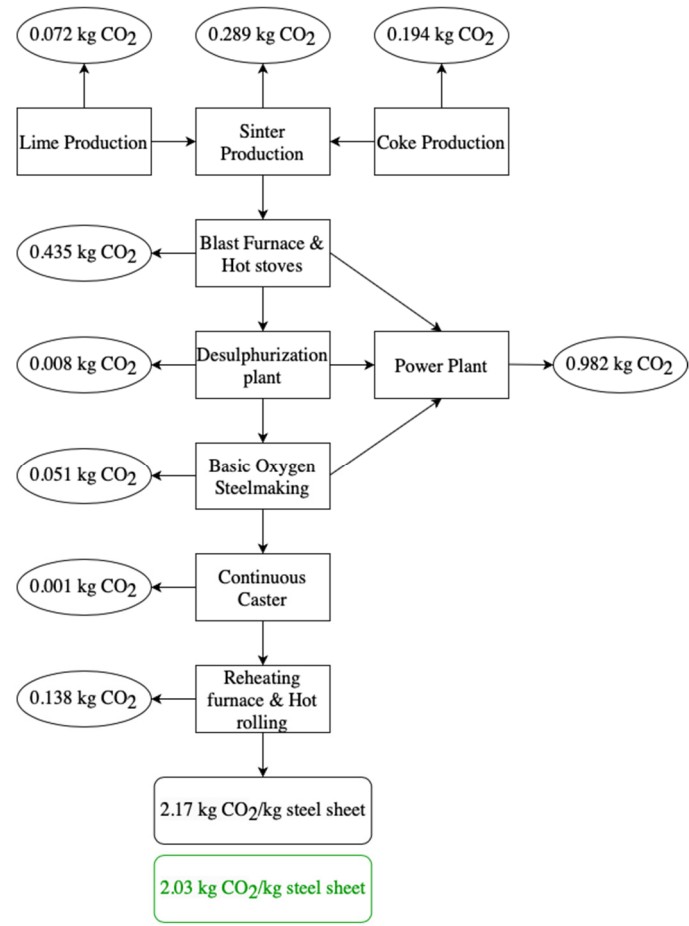

**Figure 1.** BF-BOF pathway and $CO_2$ emissions (based on—IEA, 2013 [36]) (green stands for renewable grid).

The main data were taken from the IEA 2013 [36]. Despite the availability of technologies to reduce emissions, the high cost of implementing them is a significant challenge in decarbonizing the process. The start point of the performed LCA in this study is raw-materials mining and the end point is hot-rolled sheet. The $CO_2$ emissions during sheet steelmaking through the BF-BOF route followed by hot rolling is reported to be 2.17 kg $CO_{2\,eq}$/kg steel sheet. Using renewable energy instead reduces the emissions slightly to 2.03 kg $CO_{2\,eq}$/kg steel sheet (marked green on Figure 1).

### 2.2.2. Electric Arc Furnace

The EAF method is the most widely used to produce secondary steel from steel scrap. EAF steel production contributes 28.9% of global steel production [35]. The process is commonly used for steel recycling and Directly Reduced Iron (DRI) refining. Unlike the continuous process of BF, both BOF and EAF operate in a batch mode. As the most significant electrification opportunity in the steelmaking industry, EAF production has a low carbon footprint of about 0.4 kg $CO_{2\,eq}$/kg steel sheet compared to the BF-BOF method with reported emissions of 2.17 kg $CO_{2\,eq}$/kg steel sheet, and it is easier to modify. However, its use is limited by the availability of scrap steel as a feedstock and power quality issues (e.g., it should have relatively big grid size to absorb EAF power demand surges) due to its batch operation. With the increasing penetration of intermittent renewable energy sources, the use of EAF in steelmaking production may increase.

### 2.2.3. Direct Reduction of Iron

During the DRI production process, iron ore is reduced to iron in a solid state without reaching the melting point of iron. The main reduction agent used worldwide is natural gas (NG). It is either cracked to $H_2$ and CO in the external reformer [37] and fed into the shaft reduction reactor filled with an iron oxide pellet as an iron feedstock or is cracked in the reduction reactor in contact with metallic iron [38]. Compared to the traditional BF process of producing pig iron, DRI production is more energy efficient. However, additional processing, i.e., EAF, is required to upgrade the DRI sponge iron for market use. For the current analysis, the MIDREX process was selected (Figure 2).

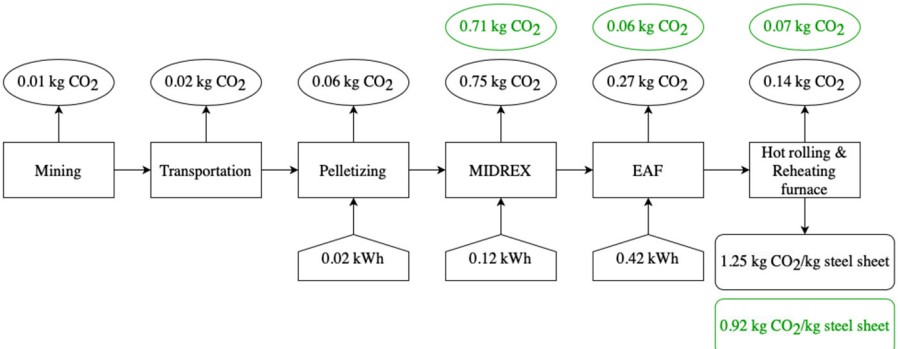

**Figure 2.** NG-DRI and EAF steelmaking (green stands for renewable grid).

During this process, the $CO_2$ emissions amount to 1.25 kg $CO_{2\,eq}$/kg sheet. It can be reduced to 0.92 kg $CO_{2\,eq}$/kg sheet when renewable energy is used instead of the traditional grid mix.

Using direct reduced iron as a feedstock can greatly improve the steel quality during EAF steelmaking process as DRI is a highly pure raw material. The combination of DRI and EAF allows for higher levels of electrification and lower emissions when low-carbon feedstocks and electricity are used. The main sources of $CO_2$ emissions are the following reduction reactions (Equations (1) and (2)) in the reactor:

$$Fe_2O_3(s) + 3\,CO(g) \rightarrow 2\,Fe(s) + 3\,CO_2(g) \tag{1}$$

$$FeO(s) + CO(g) = Fe(s) + CO_2(g) \tag{2}$$

The DRI-EAF combination has a better potential for a deep decarbonization, as the natural gas can be relatively easily enriched with hydrogen or even substituted with pure hydrogen, while the BF-BOF process faces greater difficulty in removing carbon due to embedded technological limitations—BF cannot operate without coke, which assures gas permeability in the reactor. In addition, use of $H_2$ in blast furnaces is complicated due to the overall high cost of retrofitting the existing equipment.

### 2.3. Decarbonization Methods for Steel

Along with conventional processes, decarbonized iron- and steelmaking processes such as BF-BOF with carbon capture, EAF production with 100% scrap input, and $H_2$-DRI (using hydrogen from renewable electricity sources) were studied.

#### 2.3.1. BF-BOF with Carbon Capture

The energy demand of a carbon capture unit, i.e., reboiler, $CO_2$ compression, and other auxiliaries, is 417 kWh/t $CO_2$ captured [36]. Carbon capture units utilize natural gas and electricity for running oppressions including compression, reboiler, etc. The embedded carbon dioxide in carbon capture is 0.11 t $CO_{2\,eq}$/t $CO_2$ captured with traditional electricity grid assumed in the present study, and 0.06 t $CO_{2\,eq}$/t $CO_2$ captured with assumed renewable electricity. The major sources of $CO_2$ emissions in BF-BOF-based steelmaking are the blast furnace, coking ovens, and heat/power plants. $CO_2$ from the hot stoves, steam generation plant, the coke oven batteries and the lime kiln is collected and sent to a carbon capture unit [36]. The efficiency of carbon capture is assumed to be 90% [39]. In this study, the compound used for carbon capture is aqueous mono-ethanolamine (MEA). The final emissions can be reduced from 2.174 $CO_{2\,eq}$/kg steel sheet to 0.448 kg $CO_{2\,eq}$/kg steel sheet after installing a carbon capture plant that captures 1.72 kg $CO_{2\,eq}$/kg steel and uses renewable energy as a source of electricity. Carbon dioxide is trapped and injected deep into permeable and porous geologic layers, where it is then isolated for extended periods of time.

#### 2.3.2. EAF Route

Steelmaking from steel scrap is one of the alternative processes which emits substantially less $CO_2$ compared to the traditional BF-BOF route and currently nearly 70% of steel in the USA is produced this way. Due to the limitations in copper removal from the scrap, not all types of scrap can be used for making high-end products, i.e., automotive steel sheet. Thus, to utilize 100% scrap-based production for automotive sheet, more sophisticated sorting technologies would need to be applied. The process emissions for making sheet from steel scrap are given in Table 1. The total $CO_2$ emissions from making 1 kg steel sheet from scrap steel is 0.40 kg with the traditional electricity grid and 0.18 kg $CO_{2\,eq}$/kg steel sheet with renewable electricity grid. The renewable electricity was calculated considering the coefficients used for the renewable electricity grid mix explained in the Methods section.

**Table 1.** $CO_2$ emissions for 100% scrap-based EAF steel, utilizing the traditional grid and a renewables-based grid.

| Processes | Traditional Electricity (kg $CO_{2\,eq}$/kg Steel Sheet) | Renewable Electricity (kg $CO_{2\,eq}$/kg Steel Sheet) |
|---|---|---|
| Scrap processing | 0.04 [40] | 0.04 |
| Transportation | 0.02 [40] | 0.02 |
| Direct emissions | 0.04 [8] | 0.04 |
| Emissions from electricity (indirect emissions) | 0.16 [41] | 0.01 |
| Reheating furnace | 0.06 [36] | 0.06 |
| Sheet rolling | 0.08 [40] | 0.01 |
| Total | 0.40 | 0.18 |

#### 2.3.3. $H_2$-DRI (Using Hydrogen Generated from Renewable Electricity Sources)

$H_2$-DRI is an energy-intensive steelmaking process, as significant quantities of electricity are needed for hydrogen production. The energy demand for hydrogen production is estimated as nearly 2630 kWh/t DRI [42]. To produce low $CO_2$ emission steel through $H_2$-DRI, the source of electricity used for hydrogen making should be renewable or nuclear, so that "upstream" emissions can be minimized. In the present study, the traditional grid assumption corresponds to the total emissions of 1.64 kg $CO_{2\,eq}$/t steel sheet. Use

of the decarbonized electricity would drastically reduce this number to 0.31 kg $CO_{2\,eq}$/t steel sheet.

Several studies [37–40] have been published forecasting the expected cost at scale of manufacturing iron using hydrogen direct reduction followed by EAF steelmaking (Table 2). These efforts vary somewhat in their assumptions and cost engineering approaches, resulting in variance across the final cost estimates. Within each study, the selection and analysis of the hydrolysis method (e.g., polymer membrane, alkaline, solid oxide) are quite important. For the purposes of this study, we are agnostic to the hydrolysis method and simply use cost of hydrogen as an input in the cost model.

**Table 2.** Selection of publications examining the manufacturing cost of steel through hydrogen direct reduction.

| Steel Production Scenario | Published or Forecast Year | Iron Ore Cost Assumption (USD/t) | Electricity Cost (USD/kg) | Green $H_2$ Cost (USD/kg) | Estimated Steel Mfg. Cost (USD/t) | References |
|---|---|---|---|---|---|---|
| Crude Steel ($H_2$-DRI) | 2022 | 110 | N/A | 4.3 | 660 | [43] |
| Crude Steel ($H_2$-DRI) | 2030 | 110 | N/A | 1.8 | 550 | [43] |
| Crude Steel ($H_2$-DRI) | 2040 | 110 | N/A | 1.3 | 520 | [43] |
| Steel Hot Metal ($H_2$ Injection BF-BOF) | 2021 | N/A | 0.045 | 5.57 | + 153 vs. BF-BOF | [22] |
| Steel Hot Metal (100% $H_2$-DRI) | 2021 | N/A | 0.045 | 5.57 | + 465 vs. NG-DRI | [21] |
| Steel Hot Metal ($H_2$ Injection BF-BOF) | 2021 | N/A | 0.062 | 8.88 | 360 | [44] |
| Steel Hot Metal ($H_2$ Injection BF-BOF) (Low) | 2050 | N/A | 0.025 | 4.04 | 410 | [44] |
| Steel Hot Metal ($H_2$ Injection BF-BOF) (Mid) | 2050 | N/A | 0.025 | 2.13 | 360 | [44] |
| Steel Hot Metal ($H_2$ Injection BF-BOF) (High) | 2050 | N/A | 0.025 | 1.45 | 340 | [44] |
| Liquid Steel (100% $H_2$-DRI) (Low) [a] | 2018 | 116 | 0.024 | 1.7 | 420 | [45] |
| Liquid Steel (100% $H_2$-DRI) (Mid) [a] | 2018 | 116 | 0.072 | 3.8 | 580 | [45] |
| Liquid Steel (100% $H_2$-DRI) (High) [a] | 2018 | 116 | 0.120 | 6.0 | 742 | [45] |
| Crude Steel ($H_2$-DRI) [b] | 2020 | 174 | 0.070 | 4.8 | 790 | [42] |
| Crude Steel ($H_2$-DRI) [b] | 2030 | 174 | 0.079 | 3.7 | 799 | [42] |
| Crude Steel ($H_2$-DRI) [b] | 2050 | 174 | 0.055 | 1.3 | 678 | [42] |
| Crude Steel (SOEL) [c] | 2021 | 98 | 0.089 | N/A | 1056 | [46] |
| Crude Steel (PEMEL) [c] | 2021 | 98 | 0.089 | N/A | 954 | [46] |
| Crude Steel (AEL) [c] | 2021 | 98 | 0.089 | N/A | 829 | [46] |
| Crude Steel (SOEL) [c] | 2050 | 98 | 0.063 | N/A | 628 | [46] |
| Crude Steel (PEMEL) [c] | 2050 | 98 | 0.063 | N/A | 615 | [46] |
| Crude Steel (AEL) [c] | 2050 | 98 | 0.063 | N/A | 632 | [46] |

N/A: Information not available in report or paper. [a] Assumes USD 1.15 to EUR 1 (August 2018). [b] Assumes USD 1.20 to EUR 1 (April 2021). [c] Assumes USD 1.16 to EUR 1 (October 2021); SOEL = solid oxide electrolysis; AEL = alkaline electrolysis; PEMEL = polymer electrolyte membrane electrolysis.

### 2.4. Life Cycle Analysis of Aluminum Production

2.4.1. Primary Aluminum Production

Primary aluminum production is an energy-intensive process, consuming 20.27 kWh of energy to produce 1 kg Al sheet (Figure 3).

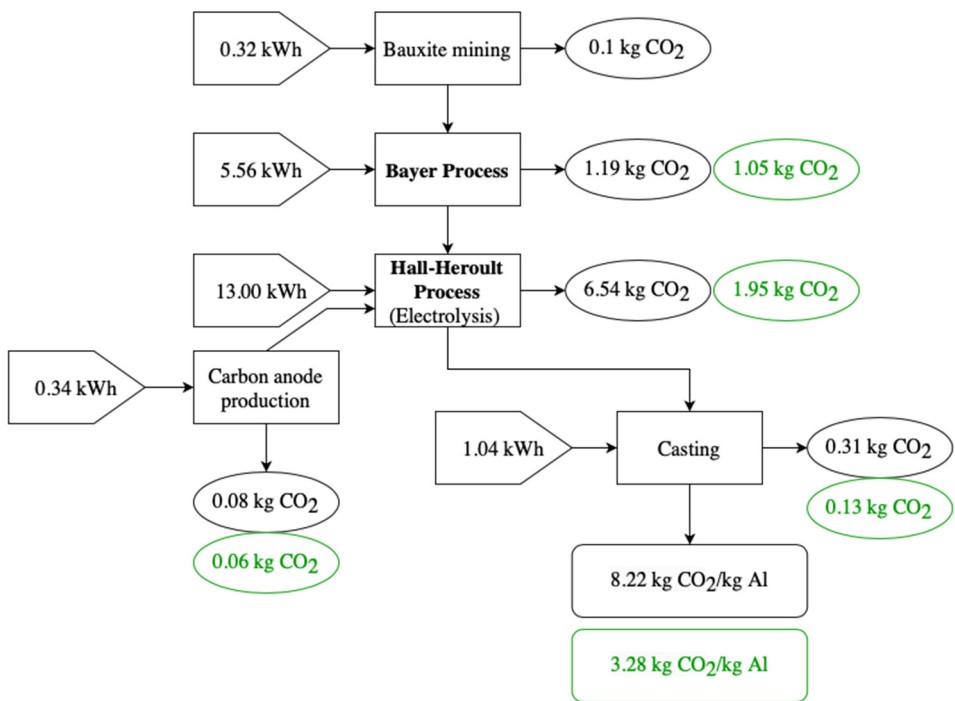

**Figure 3.** Primary Al production route (green stands for renewable grid).

To calculate the total carbon dioxide emission from process and direct emissions, the source of fuel was identified. The detailed carbon dioxide emissions are documented in Table 3.

**Table 3.** $CO_2$ emissions in primary aluminum making.

| Processes | Energy Input (kWh/kg Al Sheet) | $CO_2$ from Process Heat (kg $CO_{2\,eq}$/kg Al) | $CO_2$ from Electricity (kg $CO_{2\,eq}$/kg Al) | Direct Emissions (kg $CO_{2\,eq}$/kg Al) | Total (kg $CO_{2\,eq}$/kg Al) |
|---|---|---|---|---|---|
| Bauxite mining | 0.32 [47] | | | 0.10 | 0.10 |
| Bayer's process | 5.56 [48] | 1.04 | 0.15 | | 1.19 |
| Anode production | 0.34 [48] | 0.06 | 0.02 | | 0.08 |
| Electrolysis | 13 [48] | | 5.01 | 1.53 | 6.54 |
| Hot rolling | 1.04 [49] | 0.11 | 0.20 | | 0.31 |
| Total | 20.27 | 1.21 | 5.38 | 1.63 | 8.22 |

Bayer's process, anode production, and hot rolling use natural gas for process heating. Electricity is used as source of energy for hot rolling and electrolysis. The direct emission of $CO_2$ is through graphite anode consumption for aluminum reduction as shown in Equation (3).

$$\text{Aluminum reduction: } 2Al_2O_3(\text{non-aqueous}) + 3C(s) \rightarrow 4Al(s) + 3CO_2(g) \qquad (3)$$

### 2.4.2. Secondary Aluminum Production

Secondary aluminum takes only 2.52 kWh/kg Al sheet, which is 9% [49] of the energy needed to produce primary aluminum. It also emits less carbon dioxide than the production of primary aluminum, i.e., 0.37 kg $CO_{2\,eq}$/kg Al sheet when renewable energy is used.

### 2.5. Decarbonization Methods for Aluminum

### 2.5.1. Primary Aluminum Production and Carbon Capture

During primary aluminum production, a great amount of carbon dioxide is emitted. To reduce $CO_2$ emissions, a carbon capture unit is assumed to be installed to capture

released $CO_2$ from the Bayer process and electrolysis stack. The carbon capture unit uses MEA for post-combustion capture with heat integration. The concentration of $CO_2$ in flue gas is 4% [50]. When replacing the conventional electricity grid with renewable electricity, the final $CO_2$ emissions drop down to 1.65 kg $CO_{2\,eq}$/kg Al sheet.

2.5.2. Decarbonized Aluminum

Aluminum production emits 8.22 metric tons (t) of $CO_{2\,eq}$/t Al (see Table 4). The most carbon dioxide is emitted from the conventional electricity grid. When changing the current electricity grid to a renewable grid, additional decarbonization of the process can be achieved using inert anodes. This can eliminate direct carbon dioxide emissions from electrolysis, which produces 1.53 kg $CO_{2\,eq}$/kg Al. However, due to more potential differences in reactants, inert anodes would demand more electricity as compared to carbon ones. A carbon anode needs 13 kWh/kg aluminum, whereas an inert anode demands 16 kWh/kg [51]. The composition of the inert anode is assumed to be 51.7% NiO and 48.3% $Fe_2O_3$ [52]. It was evident from the results that the $CO_2$ emissions of 1.16 kg $CO_{2\,eq}$/kg Al were obtained from a route when renewable energy is used as source of energy for process heat in the Bayer process, and the inert anode is used for electrolysis. A comparison between conventional and decarbonized aluminum-making is shown in Table 4.

**Table 4.** Comparison of decarbonized aluminum with conventional aluminum (primary).

| Processes | Conventional Aluminum (kg $CO_{2\,eq}$/kg Al Sheet) | Decarbonized Aluminum (kg $CO_{2\,eq}$/kg Al Sheet) | Technologies |
|---|---|---|---|
| Bauxite mining | 0.10 | 0.10 | No Change |
| Bayer process | 1.19 | 0.18 | Thermal Energy from Renewable Electricity |
| Anode production | 0.08 | 0.25 | Inert Anodes |
| Electrolysis | 6.54 | 0.51 | Inert Anodes |
| Hot rolling | 0.31 | 0.13 | Renewable electricity |
| Total | 8.22 | 1.16 | |

*2.6. LCA for Glass Fiber Composites*

Mass and energy balance are adapted from the data reported by Dai et al. [53] and Hill et al. [54] (Figure 4 and Table 5).

The energy requirement and $CO_2$ emissions for epoxy resin are taken from the LCA database reported by Hill et al. [54]. Fuel burnt in process heat is natural gas. When batch materials melt, they undergo decomposition and release $CO_2$ through reactions according to Equations (4)–(6). The amount of $CO_2$ emitted is calculated stoichiometrically.

$$CaCO_3 \rightarrow CaO + CO_2(g) \tag{4}$$

$$Na_2CO_3 \rightarrow Na_2O + CO_2(g) \tag{5}$$

$$CaMg(CO_3)_2 \rightarrow CaO + MgO + 2CO_2(g) \tag{6}$$

To use composite sheets, it should be reinforced with an electrical glass (E-glass). There are several types of composite manufacturing techniques depending on the final product. To make composites for automobile parts, the following molding processes are used: additive manufacturing, sheet molding compound or compression molding, filament winding, vacuum infusion, continuous process injection, and over molding. Resin is needed to apply to the glass fabric's surface to produce the composite. The automotive sector primarily uses five main types of resins to manufacture carbon-fiber-based composites. Thermosetting (e.g., epoxy) and thermoplastic resins are often used in the automotive

industry for automobile bodies, while phenolic and bismaleimide (BMI) cyanate resins are used for the engine, engine compartment, and gear box. For this study, compression molding is considered. It uses electricity as an energy source. The polymer considered is epoxy resin. The ratio of resin to fiber used is 50:50 [53]. Resin production followed by refining and melting are the most energy-intensive process steps. $CO_2$ emissions are 3.48 kg $CO_2$ $_{eq}$/kg GFRC. Table 5 gives process-wise $CO_2$ emissions in production of GFRC. For refining and melting, natural gas is used as an energy source.

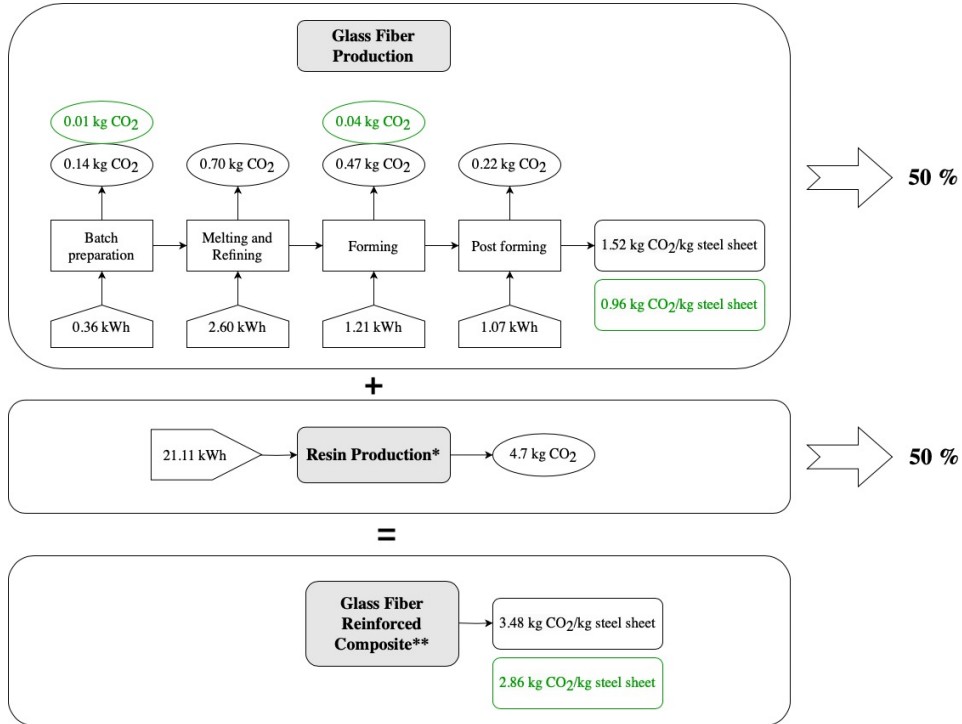

**Figure 4.** A schematic overview of glass-fiber-reinforced composites production (green stands for renewable grid). * The multistage resin production process is described by Hill et al. [54]. ** Considering fabrication by SMC.

**Table 5.** $CO_2$ emissions from production of 1 kg glass-fiber-reinforced composites.

| Process | Energy (kWh/ kg GFRC) | $CO_2$ from Process Heat (kg $CO_2$ $_{eq}$/ kg GFRC) | $CO_2$ from Electricity (kg $CO_2$ $_{eq}$/ kg GFRC) | Direct Emissions (kg CO2 eq/ kg GFRC)) | Total (kg $CO_2$ $_{eq}$/ kg GFRC) |
|---|---|---|---|---|---|
| Batch preparation | 0.36 [53] | | 0.14 | | 0.14 |
| Melting and refining | 2.60 [53] | 0.53 | | 0.17 | 0.70 |
| Forming | 1.21 [53] | | 0.47 | | 0.47 |
| Post-forming | 1.07 [53] | 0.22 | | | 0.22 |
| Total | 5.23 | | | | 1.52 |
| Epoxy resin production | 21.11 [54] | 4.7 | | | 4.70 |
| **Resin: reinforcement = 50:50** | | | | | |
| Fabrication using sheet molding compound (SMC) | 0.97 [55] | | | 0.37 | 0.37 |
| Materials used | 13.17 | | | | 3.11 |
| Total | 14.15 | | | | 3.48 |

### 2.7. LCA for CFRC

Carbon fiber is mainly produced from polyacrylonitrile (PAN) and rayon or petroleum pitch. The manufacturing process of carbon-fiber-reinforced composite production is shown in Figure 5.

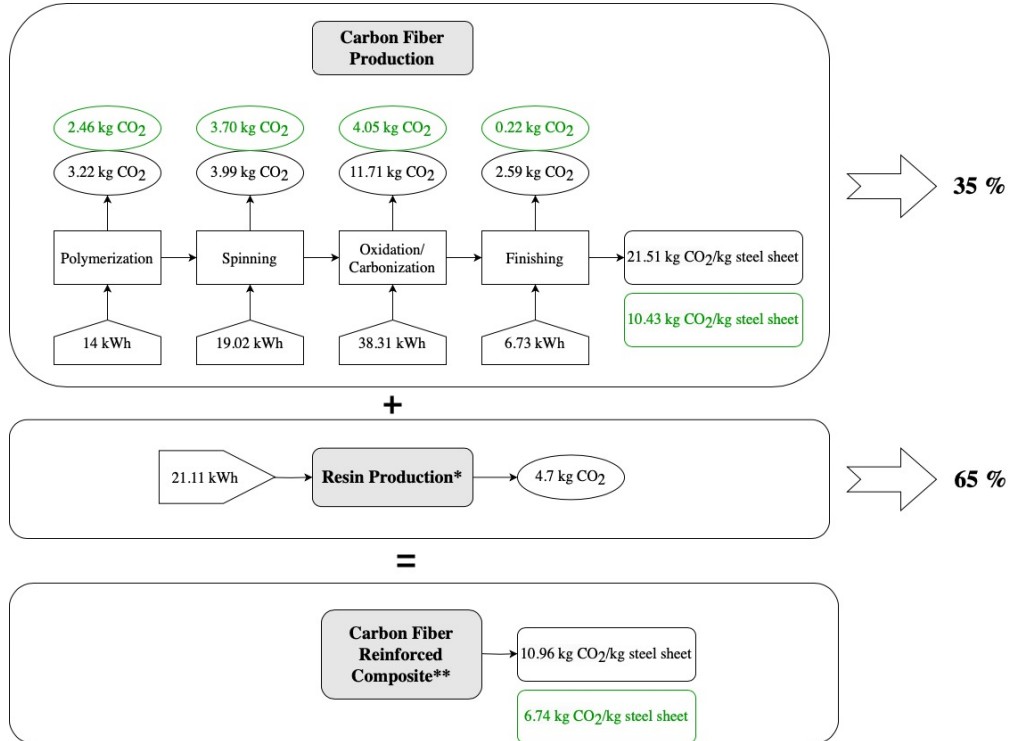

**Figure 5.** Schematic overview of carbon-fiber-reinforced composite production. * Multistage resin production process is described by Hill et al. [54]. ** Considering fabrication by SMC.

The energy required for the production of carbon fiber is taken from the report published by Department of Energy, United States (See Table 6) [55].

**Table 6.** Energy requirement for carbon-fiber-reinforced composite production [55,56].

| Process | Energy (kWh/kg) | Natural Gas (%) | Electric (%) |
|---|---|---|---|
| Polymerization | 14.00 | 84.70 | 15.30 |
| Spinning | 19.02 | 95.70 | 4.30 |
| Oxidation/Carbonization * | 38.31 | 43.40 | 56.60 |
| Finishing | 6.73 | 0.00 | 100 |
| Resin production | 21.11 | 91.90 | 8.10 |
| Fabrication by SMC | 0.97 | 0.00 | 100 |

* Used in TEA.

CFRC produces total emissions of 10.96 kg $CO_{2\,eq}$/kg CFRC with the major contributors being oxidation of carbon fiber followed by resin production. The detailed $CO_2$ emissions are shown in Table 7. As with GFRC, carbon fiber is combined with a polymeric material to make a robust composite. In this study, epoxy resin is considered as the resin and the ratio of fiber to resin is 35:65 [57].

**Table 7.** $CO_2$ emissions in production of 1 kg carbon-fiber-reinforced composite.

| Processes | Emissions from Process Heat (kg $CO_{2\ eq}$/kg CFRC) | Emissions from Electricity (kg $CO_{2\ eq}$/kg CFRC) | Total Emissions (kg $CO_{2\ eq}$/kg CFRC) |
|---|---|---|---|
| Polymerization | 2.39 | 0.82 | 3.22 |
| Spinning | 3.68 | 0.31 | 3.99 |
| Oxidation/Carbonization | 3.36 | 8.35 | 11.71 |
| Finishing | 0.00 | 2.59 | 2.59 |
| Total | 9.43 | 12.08 | 21.51 |
| Epoxy resin production | 4.70 | | 4.70 |
| **Resin: reinforcement = 65:35** | | | |
| Fabrication by SMC | | 0.37 | 0.37 |
| Materials used | | | 10.58 |
| Total | | | 10.96 |

*2.8. Manufacturing Cost Analysis*

Cost analysis was performed to determine manufacturing cost per tonne of the previously described materials and processes. In general, the cost is determined by estimation via process modeling or from available data in literature factored by the quantity of material and energy inputs to the process. These inputs are harmonized with the previously described efforts that estimate Global Warming Potential. Capital expenses and labor requirements per tonne are estimated using assumptions found in the literature and in Supplemental Information Section S4. These are multiplied by the 2019 cost per unit value found in the literature or commodity market bulletins. Details of all cost models can be found in the Supplementary Information (See Sections S1–S5) [20,58–84].

Costs found in the literature are assumed to be accurate as of the date of publication. They are adjusted to 2019 US Dollars using historical exchange rates (if necessary) and the US Consumer Price Index for all Urban Consumers (CPI-U) [84].

The year 2019 was chosen as the basis for analysis as it is representative of manufacturing costs in the years before the global COVID-19 pandemic and Ukraine conflict. Although labor, feedstock, and energy prices always change, we assert that the 2019 prices cited in this work are representative of long-term, stable prices.

Between early-2019 and mid-2022, many input (e.g., ore, energy) and finished steel prices climbed dramatically due to supply chain volatility. Current commodity values seem to be returning to values somewhat similar to our cost assumptions. For example, we assume iron ore prices of USD 107/t, the average for 2019. Iron ore prices peaked at USD 220/t in May 2021, but have returned to USD 112/t as of June 2023 [85]. Likewise, we assume natural gas prices of USD 3.84/million cubic feet (MCF, Miami, FL, USA), which was the 2019 average industrial price. June 2023 US Industrial prices are USD 3.64/MCF after reaching peaks of USD 9.95/MCF in September 2022.

In steelmaking and aluminum-making models, the cost of ores is a key contribution to total manufactured cost. The values used in this paper assume one grade and therefore price for iron ore and bauxite. This may not be the case, but suitability of ores for a given process and analysis of ore grade prices was considered beyond the scope of the research. All total costs include a carbon tax of USD 80/t $CO_2$. Therefore, the net $CO_2$ emissions in tonnes per tonne of material produced is multiplied by USD 80 and added to the manufacturing cost to result in a "Total Cost." Sensitivity analysis to this carbon tax was applied using scenarios of USD 0/t $CO_2$ and USD 160/t $CO_2$. No carbon tax is currently implemented in the United States, but in December 2022, the European Parliament reached an agreement to implement a Carbon Border Adjustment Mechanism (CBAM) [86]. The price of EU Carbon Permits has fluctuated around USD 80/t in recent years; therefore, this value was selected as a proxy for a carbon tax. In practice, manufacturers will likely implement $CO_2$ emissions

mitigation practices, which are less expensive per tonne of $CO_2$ abated than the price of the credits.

Cost analysis of glass fiber composites and carbon fiber composites presents a challenge that is beyond the scope of this paper. The concept of producing a sheet is intrinsically tied to the volume produced, as composite part manufacturing processes tend to be batch processes. As such, a high-volume sheet product will have lower per part cost for labor, capital depreciation, overheads, etc. This contrasts with steel and aluminum sheet production, which are highly automated continuous or semi-continuous processes. To "sidestep" this difficulty and provide guidance for policymakers, design engineers, and materials manufacturers we have attempted to simplify this challenge. For composite parts, the high variance described above is simplified to one parameter in our cost models—"non-materials costs." In-depth lightweighting studies for the automotive industry have indicated that non-materials-based costs can be approximately 50% of total part costs [87,88]. Therefore, we have made this assumption to provide an "apples to apples" comparison, which incorporates the complex challenges of material selection in diverse automotive applications while addressing the issue of raw material GWP, manufacturing costs, and associated carbon taxes. This method provides a framework for decisionmakers with detailed part manufacturing costs to easily estimate the impacts of carbon taxes on their materials selection choices.

Carbon Capture

The economic perspective of carbon capture, storage, and transport was studied with respect to steel and aluminum plants. The difference in cost is observed due to differences in $CO_2$ concentrations in flue gas of steel and aluminum plants. Generally, capital expenses are inversely proportional to the concentration of $CO_2$ in the flue gas. Along with the concentration difference, there are other factors that affect the cost of carbon capture, so it was decided to use different costs for different industries. The cost of transporting $CO_2$ varies depending on the mode of transportation (e.g., pipelines vs. ships) quantity of $CO_2$ transported), the distance to the $CO_2$ storage facility, the monitoring and regulatory requirements, including any policy barriers and incentives, the cost structures related to financing, capital, and labor, and the $CO_2$ source and whether or to what extent it is pressurized or purified before transporting. These factors all differ by area because of regional variations. $CO_2$ storage costs are influenced by three key causes of variation: (1) geology properties; (2) scale (amount of $CO_2$ stored); and (3) monitoring, financial, and other assumptions.

Table 8 provides a summary of carbon capture, transport, and storage costs and assumptions from a selection of published works. For this study the average cost of USD 90/t carbon dioxide for $CO_2$ capture was assumed for steel processes and USD 127/t $CO_2$ for aluminum processes. These values are the average costs for the relevant process inflation adjusted to 2019 using the US CPI-U [84]. It was assumed that 3.2 Mtpa $CO_2$ is transported over 100 miles for onshore storage. Again, using inflation-adjusted assumptions from Table 8, we assume cost of compression, transport, and storage has a levelized cost of USD 10/t $CO_2$. Therefore, total cost for steel carbon capture and storage is assumed to be USD 100/t $CO_2$ and USD 137/t $CO_2$ for steel and aluminum, respectively.

**Table 8.** Carbon capture and storage cost for steel and aluminum plants.

| Scenario | Cost (USD/t CO$_2$ | Published Year/Forecast Year | Assumptions | References |
|---|---|---|---|---|
| Carbon capture for steel | 68.7 | 2013 | Post-combustion capture | [89,90] |
| Carbon capture for steel | 65.1–119.2 | 2013 | Post-combustion capture | [91] |
| Carbon capture for steel | 78.5 | 2011 | Post-combustion capture with MEA of blast furnace flue gas | [92] |
| Carbon capture for steel | 104.21 | 2020 | N/A | [93] |
| Carbon capture for aluminum (4%) | 123.51 | 2013 | MEA-based carbon capture, concentration of CO$_2$ in flue gas is 4% | [50] |
| Carbon capture for aluminum (7%) | 115.84 | 2013 | MEA-based carbon capture, concentration of CO$_2$ in flue gas is 7% | [50] |
| Carbon capture for aluminum (10%) | 110.52 | 2013 | MEA-based carbon capture, concentration of CO$_2$ in flue gas is 10% | [50] |
| Transport | 3.1 | 2014 | 3.2 Mtpa CO$_2$ over 100 miles, onshore | [94] |
| Transport | 4.9 | 2018 | 3 Mtpa CO$_2$ over 155 miles, onshore pipeline | [95] |
| Storage | 4.32 | 2015 | Depleted O&G Field—reusing wells onshore | [31] |
| Storage | 5.76 | 2015 | Depleted O&G Field—no reusing wells onshore | [31] |
| Storage | 7.2 | 2015 | Saline formations onshore | [31] |
| Storage | 8.64 | 2015 | Depleted O&G Field—reusing wells offshore | [31] |
| Storage | 14.4 | 2015 | Depleted O&G Field—no reusing wells offshore | [31] |
| Storage | 20.16 | 2015 | Saline formations offshore | [31] |

*2.9. Normalization Using Material Properties*

In this study, automotive sheet was chosen as the baseline product for comparison. To produce automobiles with improved fuel economy or battery range, the automotive industry's current advancements are encouraging the usage of lightweight materials including aluminum, glass-fiber-reinforced composites, and carbon-fiber-reinforced composites. Due to the increased customer appeal and compliance with legal requirements, many businesses are attempting to reduce the weight of automobiles. A potential engineering solution in this regard is aluminum, with a density around one-third that of steel and high-strength alloys that fulfill the torsion and stiffness requirements for automotive components. To make strong, stiff, and lightweight materials, FRP-composites often blend high-strength, high-stiffness fibers with low-density matrix materials. These characteristics, together with improved moldability, a favorable strength to weight ratio, and corrosion resistance, provide reinforced composites an edge over steel in the automobile sector.

Based on the bending stiffness of sheets composed of various materials, weight reduction is determined. To compare the CO$_2$ emissions and cost of steel with other materials, the equivalent factor is calculated using weight reduction.

For example, to compare steel and aluminum, Equations (7)–(11) were applied.

Bending stiffness is determined by [96]:

$$K \propto E \cdot t^3 \tag{7}$$

where K—stiffness (N/m); E—elastic modulus (GPa); t—sheet thickness (cm).

$E_{Al}$ = 71 GPa; $E_{steel}$ = 207 GPa; $\rho_{Al}$ = 2.8 g/cm$^3$; $\rho_{steel}$ = 7.8 g/cm$^3$.

To achieve the same stiffness,

$$K_{steel} = K_{Al}$$

$$\frac{t_{Al}}{t_{steel}} = \left[ \frac{E_{steel}}{E_{Al}} \right]^{\frac{1}{3}} \tag{8}$$

$$\frac{E_{steel}}{E_{Al}} = 3 \tag{9}$$

$$\frac{t_{Al}}{t_{steel}} = 1.44 \tag{10}$$

$$\text{weight savings for Al} = \left(1 - \left[\frac{(\rho_{Al} * 1.44)}{\rho_{steel}}\right]\right) * 100 = 50\% \qquad (11)$$

(In this work, the relationship of bending stiffness vs. thickness is taken as $K \propto E \cdot t^3$. The bending stiffness should not be considered proportional to cube of thickness for every auto component. That means that the weight saving is not the same in every case. In some cases, there may even be weight gain as well).

Similarly, weight reduction for GFRC and CFRC were calculated. The results are given in Table 9.

**Table 9.** Properties and results of material mass normalization. Functionally equivalent masses of sheets are derived using the concept of stiffness under bending.

| Material | E: Elastic Modulus (GPa) | ρ: Density (g/cc) | Equivalent Factor (Material/kg Steel) | Weight Savings(%) |
|---|---|---|---|---|
| Steel | 207 [97] | 7.8 [97] | 1 | NA |
| Aluminum | 71 [97] | 2.8 [97] | 0.5 | 50 |
| Glass-Fiber-Reinforced Composite | 41 [98] | 1.9 [98] | 0.4 | 60 |
| Carbon-Fiber-Reinforced Composite | 95.5 [99] | 1.4 [99] | 0.23 | 77 |

## 3. Results

### 3.1. Comparison of Materials' Global Warming Potentials, Raw and Normalized

When using a renewable electricity grid, the lowest GWP per kg of material was determined to be steel sheet made through EAF using 100% scrap input. When considering the weight reduction factor, secondary aluminum-making gives an equivalent normalized GWP with 0.18 t $CO_{2\,eq}$/t steel sheet eq. The process that has the highest GWP when using renewables-based electricity is production of CFRC with 6.74 t $CO_{2\,eq}$/t material, but when considered the weight reduction potential, emissions drop to 1.55 t $CO_{2\,eq}$/t of material.

Tables 10 and 11 summarize the global warming potential of the different processes when the traditional and renewable electricity grids are used, respectively.

**Table 10.** Global Warming Potential and Normalized Global Warming Potential of studied materials and processes.

| | Traditional Electricity Grid | | | |
|---|---|---|---|---|
| Process/Material | CCS (t $CO_2$ Captured) | Global Warming Potential (GWP) (t $CO_{2\,eq}$/t Material) | Stiffness Normalization Factor (t Material/t Steel) | Normalized GWP (nGWP) (t $CO_{2\,eq}$/t Steel Sheet eq) |
| BF-BOF Steel | 0 | 2.17 | 1.00 | 2.17 |
| Electric Arc Furnace Steel (100% scrap) | 0 | 0.40 | 1.00 | 0.40 |
| Natural Gas DRI-EAF Steel | 0 | 1.25 | 1.00 | 1.25 |
| Hydrogen DRI-EAF Steel | 0 | 1.64 | 1.00 | 1.64 |
| BF-BOF Steel with CCS | 1.42 | 0.70 | 1.00 | 0.70 |
| NG-DRI-EAF Steel with CCS | 0.64 | 0.68 | 1.00 | 0.68 |
| Primary Aluminum | 0 | 8.22 | 0.50 | 4.11 |
| Secondary Aluminum (100% scrap) | 0 | 0.45 | 0.50 | 0.22 |
| Primary Aluminum with CCS | 2.32 | 6.84 | 0.50 | 3.42 |
| Decarbonized Aluminum | 0 | 8.96 | 0.50 | 4.48 |
| Glass Fiber Composites (Epoxy) | 0 | 3.48 | 0.40 | 1.39 |
| Carbon Fiber Composites (Epoxy) | 0 | 10.96 | 0.23 | 2.52 |

**Table 11.** Global Warming Potential and Normalized Global Warming Potential of studied materials and processes.

| Process/Material | Renewables-Based Electricity Grid | | | |
| --- | --- | --- | --- | --- |
| | CCS (t $CO_2$ Captured) | Global Warming Potential (GWP) (t $CO_{2\,eq}$/t Material) | Stiffness Normalization Factor (t Material/t Steel) | Normalized GWP (nGWP) (t $CO_{2\,eq}$/t Steel Sheet eq) |
| BF-BOF Steel | 0 | 2.03 | 1.00 | 2.03 |
| Electric Arc Furnace Steel (from 100% scrap) | 0 | 0.18 | 1.00 | 0.18 |
| Natural Gas DRI-EAF Steel | 0 | 0.92 | 1.00 | 0.92 |
| Hydrogen DRI-EAF Steel | 0 | 0.31 | 1.00 | 0.31 |
| BF-BOF Steel with CCS | 1.42 | 0.43 | 1.00 | 0.43 |
| NG-DRI-EAF Steel with CCS | 0.64 | 0.33 | 1.00 | 0.33 |
| Primary Aluminum | 0 | 3.28 | 0.50 | 1.64 |
| Secondary Aluminum (from 100% scrap) | 0 | 0.37 | 0.50 | 0.18 |
| Primary Aluminum with CCS | 2.32 | 1.65 | 0.50 | 0.83 |
| Decarbonized Aluminum | 0 | 1.16 | 0.50 | 0.58 |
| Glass Fiber Composites (Epoxy) | 0 | 2.86 | 0.40 | 1.15 |
| Carbon Fiber Composites (Epoxy) | 0 | 6.74 | 0.23 | 1.55 |

These tables give the GWP in t $CO_{2\,eq}$/t material as well as in t $CO_{2\,eq}$/t steel sheet $_{eq}$. The normalizing factor plays a major role in comparing different materials such as steel, aluminum, glass fiber composites, and carbon fiber composites as they have different physical properties and are used in different proportions for the same applications. When using the traditional electricity grid, the most $CO_2$-intensive material is CFRC, i.e., 10.96 t $CO_{2\,eq}$/t material and the least $CO_2$-intensive process is steel sheet production through EAF using 100% scrap, emitting 0.40 t $CO_{2\,eq}$/t material. After considering the normalization factor, the most $CO_2$ is emitted from decarbonized aluminum-making, which is 4.48 t $CO_{2\,eq}$/t steel sheet $_{eq}$ and the least amount of $CO_2$ is emitted from secondary aluminum-making with 0.22 t $CO_{2\,eq}$/t steel sheet $_{eq}$.

If assuming the weight-reduction factor, BF-BOF steel gives the highest emissions of 2.03 t $CO_{2\,eq}$/t steel sheet $_{eq}$.

Steel and aluminum manufacturers do not always have the option to manufacture their products from 100% recycled scrap. As discussed previously, ore-based metals are usually required to dilute residuals and impurities to produce high-quality products such as automotive sheets. Second, in a growing economy, available scrap may not be sufficient to meet total market demand. This is especially true if recycling supply chains are not sufficiently developed. Therefore, it is important to note that the results of the analysis indicate that the lowest GWPs of processes for "new" or ore-based materials are natural gas DRI-EAF steel with CCS (traditional grid) and hydrogen DRI-EAF (renewable grid). Hydrogen DRI is currently envisioned as a key technology for the steel industry to reduce greenhouse gas emissions, but it will be essential to pair this technology with an improvement in overall grid emissions. The effectiveness of $H_2$-DRI and decarbonized aluminum efforts will be predicated on massive development of renewable electricity generation capacity. This is not the case for CCS-based technologies, either NG-DRI or BF-BOF. Both fossil fuel-based technologies show tremendous opportunity in the near term to reduce steel emissions without requiring massive changes to the grid.

### 3.2. Comparison of Materials' Manufacturing Costs, and Potential Assessed Carbon Tariffs

Figures 6 and 7 display the results of the manufacturing cost analysis, assessed carbon tax (or tariff), and associated GWP of the relevant materials and processes when assuming the traditional grid. As described in the Methods section, the baseline carbon tax for each material is USD 80/t $CO_2$ eq. A precise breakdown of cost calculations can be found in the

Supplemental Information, Section S1. Overall, it should be observed that carbon taxes add substantially to the total cost of the products on both a raw basis (per kg of material) and a performance-adjusted basis (per kg of steel sheet equivalent).

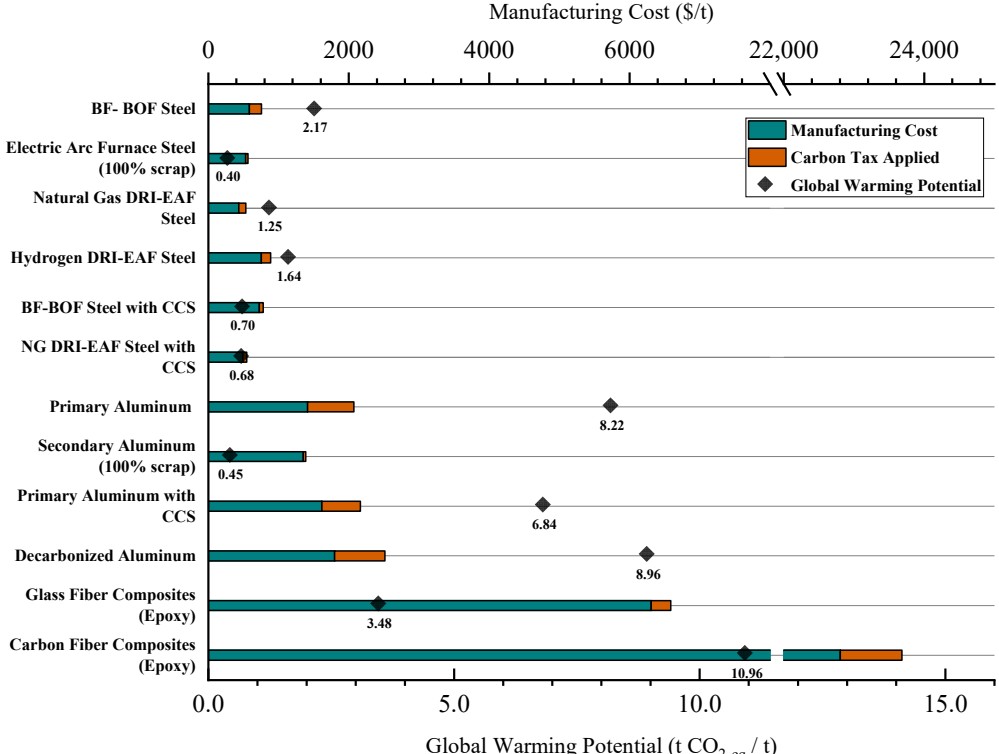

**Figure 6.** Manufacturing cost, carbon tax applied, and Global Warming Potential for the studied materials and processes (traditional grid). These are displayed per tonne of finished sheet of the material at factory gate. The carbon tax applied is USD 80 per tonne of $CO_2$ equivalent emissions as measured using the cradle-to-gate LCA method. The LCA assumes a traditional electricity grid with a GWP of 0.385 kg $CO_{2\,eq}$/kWh. Note the break in the manufacturing cost axis as carbon fiber composite is more expensive per tonne than other materials.

For the traditional grid scenario, the application of carbon taxes does not affect the cost rank order of any steel process relative to another steel process. We speculate that in the future some customers may prefer (and be willing to pay for) lower GWP products. However, customers seeking low-cost steel sheets in a carbon tax regime will not be incentivized to change behavior because of the carbon tax in this traditional grid scenario.

Aluminum is particularly impacted, but the reality is that many aluminum manufacturers are already co-located with low-cost/low-carbon hydroelectric grids. Aluminum manufacturers that are drawing from fossil-fuel-heavy energy grids will clearly be heavily impacted by carbon taxes. These manufactures should investigate green certification using new physical renewable sources (e.g., via Power Purchase Agreements) or a synthetic renewable grid via financial instruments such as Renewable Energy Certificates. If these are unavailable, significant use of scrap is heavily incentivized by the implementation of carbon taxes. In the total balance of emissions from Al production, only a small share can be captured by CCUS (from the Bayer process and carbon anodes), while indirect emissions from the grid remain unaffected.

Our assumptions on the decarbonization of aluminum production were impacting the direct emissions from the Hall–Héroult process, e.g., the $CO_2$ emissions from carbon anodes of about 1.53 t $CO_{2\,eq}$/t Al. However, usage of inert anodes leads to higher electricity consumption. Therefore, under the assumption of the same grid GHG footprint for both traditional and decarbonized Al production, the effect of inert anodes' $CO_2$ saving is offset

by the increasing indirect $CO_2$ emissions from the power grid. In other words, inert anode implementation makes sense only for the facilities utilizing low-$CO_2$ energy sources of electricity (hydro, wind, solar, nuclear).

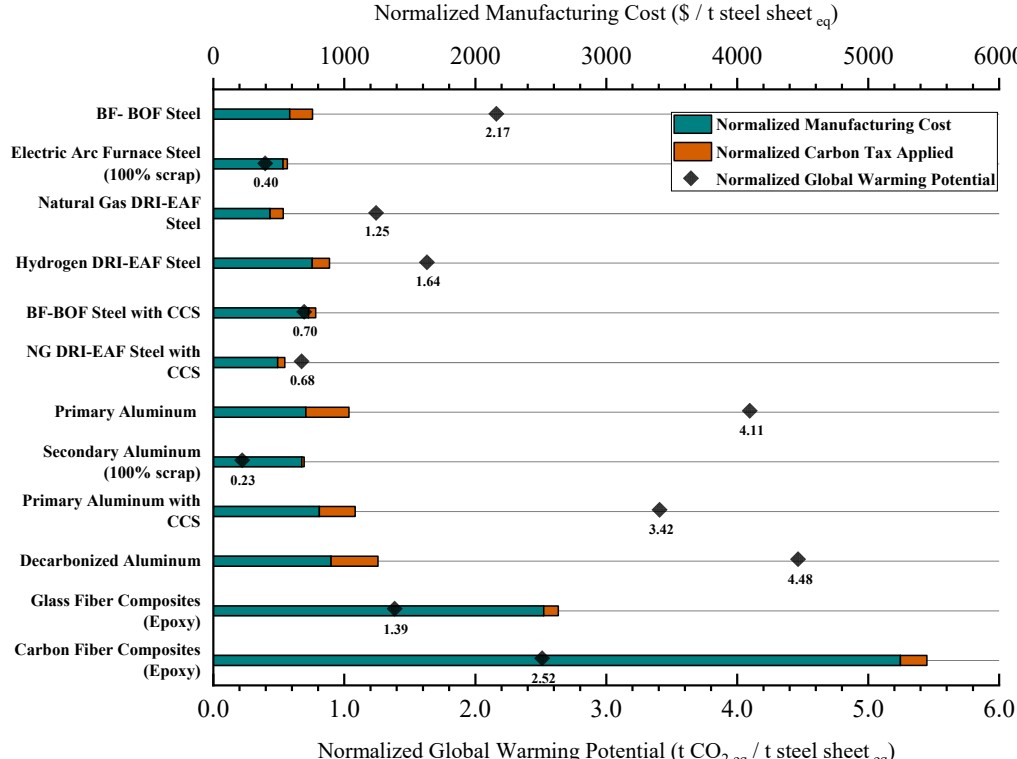

**Figure 7.** Normalized manufacturing cost, carbon tax, and Global Warming Potential for the studied materials and processes (traditional grid). The masses of the materials are normalized using elastic modulus and density as described in Methods to result in functionally equivalent sheet products. The LCA assumes a traditional electricity grid with GWP of 0.385 kg $CO_{2\,eq}$/kWh.

Glass fiber and carbon fiber composites are not as heavily impacted on a percentage basis as aluminum. Lightweighting efforts by vehicle manufacturers will likely not be impeded by carbon taxes. This study has not attempted to quantify manufacturing changes necessitated by evolving from steel parts to aluminum or composite parts. Nor does the study analyze Life Cycle Impacts of end use, such as better gas mileage achieved by lightweight material substitution.

Figures 8 and 9 illustrate the results of the manufacturing cost analysis, assessed carbon tax, and associated GWP of the materials and processes when assuming a renewables-based grid.

The products which require significant amounts of electrical energy show improved total cost thanks to reduced carbon taxes. Products with high direct emissions such as BF-BOF steel and primary aluminum do not benefit greatly from the lower emissions upstream in power generation. Our analysis indicates that these manufacturers have few options in a moderate carbon tax regime other than to integrate more scrap into their operations.

Our assumption of CCS charges for aluminum (USD 137/t $CO_2$) is greater than the carbon tax (USD 80/t $CO_2$), so a "bolt-on" CCS solution may not make sense. The scenario that could make sense is if certain unit operations have $CO_2$ emissions that can be easily captured at costs lower than the carbon taxes. Likewise, the assumption for CCS charges for BF-BOF steel (USD 100/t $CO_2$) is greater than the carbon tax. Some studies have indicated that CCS costs could be lower than USD 80/t $CO_2$ for some unit operations, so investment in CCS assets may reduce total costs [100].

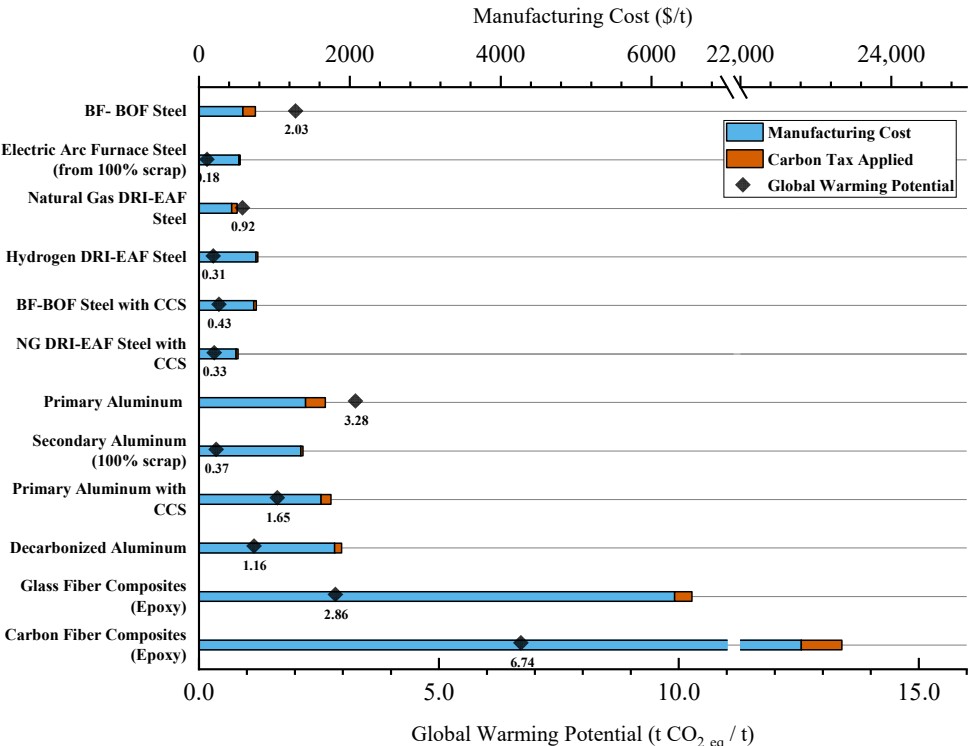

**Figure 8.** Manufacturing cost, carbon tax applied, and Global Warming Potential for the studied materials and processes (renewables-based grid). These are displayed per tonne of finished sheet of the material at factory gate. The carbon tax applied is USD 80 per tonne of $CO_2$ equivalent emissions as measured using the cradle-to-gate LCA method. The LCA assumes a renewables-based electricity grid with a GWP of 0.038 kg $CO_{2\,eq}$/kWh. Note the break in the manufacturing cost axis as carbon fiber composite is more expensive per tonne than other materials.

In the renewable grid scenario, glass fiber composites and carbon fiber composites remain more expensive on a performance-normalized basis than steel and aluminum products. We note that end products (e.g., automotive parts) are manufactured in dramatically different volumes with huge variation in labor, capital, engineering, and overhead. Cost per part analysis is beyond the scope of this study, but this analysis does seem to indicate that carbon taxes will not impact materials selection decisions. If a specific composite material provides the required performance and economics, the addition of carbon taxes will likely not impact the design engineer's material choice.

Figures 10 and 11 illustrate the sensitivity of total costs (manufacturing cost plus carbon tax) using scenario assumptions of USD 0/t $CO_2$, USD 80/t $CO_2$ (baseline), and USD 160/t $CO_2$. Products and processes with a high GWP are heavily impacted in the USD 160/t scenario. As stated previously, $H_2$-DRI as a carbon-abating process is predicated on reducing upstream emissions via a renewable grid. In the traditional grid scenario, $H_2$-DRI never provides economic value relative to NG-DRI with CCS. Aluminum is heavily impacted by doubling assumed carbon taxes. The caveat described previously still applies, which is that manufacturers will likely need to demonstrate the use of renewable electricity (either directly or by credits) to remain cost competitive.

Figure 12 examines the sensitivity of total cost of manufacturing one tonne of steel via the DRI-EAF route (including USD 80/t $CO_2$ carbon taxes). The NG scenario includes CCUS. The cost of $H_2$-DRI-EAF steel increases as the assumed cost of hydrogen increases, crossing over several breakeven points for scenarios involving natural gas. The baseline cases discussed previously assume feedstock costs of USD 4.5/kg for $H_2$-DRI and USD 3.8/MCF for natural gas.

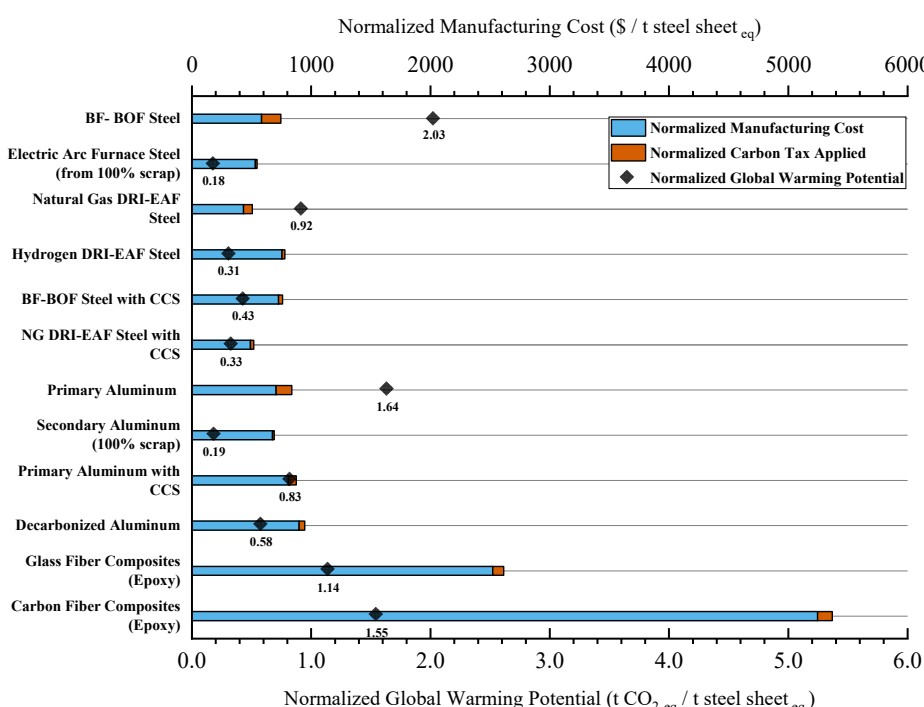

**Figure 9.** Normalized manufacturing cost, carbon tax, and Global Warming Potential for the studied materials and processes (renewables-based grid). The masses of the materials are normalized using elastic modulus and density as described in Methods to result in functionally equivalent sheet products. The LCA assumes a renewables-based electricity grid with GWP of 0.038 kg $CO_{2\,eq}$/kWh.

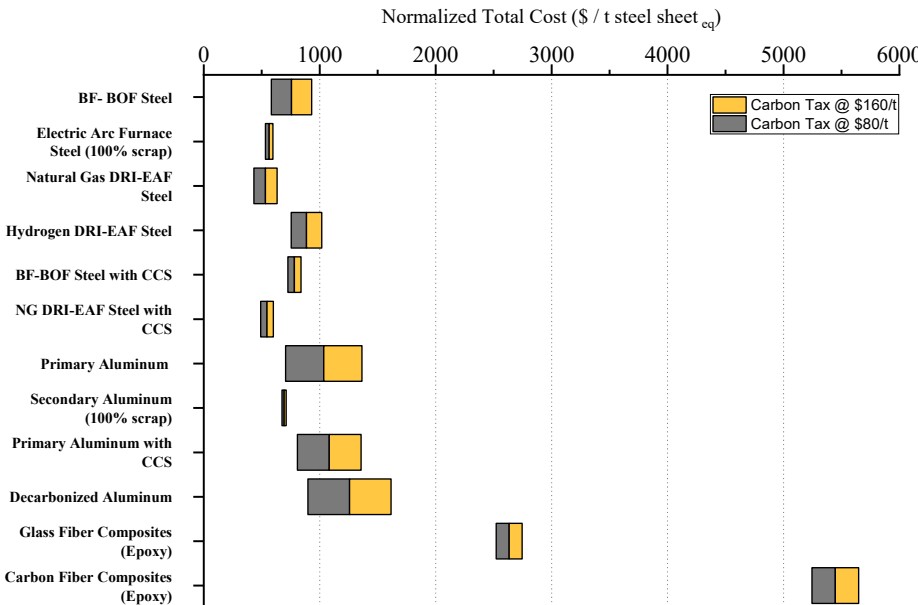

**Figure 10.** Sensitivity analysis of carbon tax on total manufacturing costs (USD/t steel sheet $_{eq}$) (traditional grid). Total cost per tonne is defined as the total manufacturing cost of the material plus applied carbon taxes. The costs are normalized according to the equivalent mass method for a sheet product as described in Methods. The midline of each bar depicts the base case of a USD 80/t $CO_2$ tax applied on carbon emissions. The left boundary of the gray bar depicts the manufacturing cost with no carbon tax applied. The right boundary of the yellow bar depicts the application of a carbon tax equal to USD 160/t $CO_2$. The LCA assumes a traditional electricity grid with a GWP of 0.385 kg $CO_{2\,eq}$/kWh.

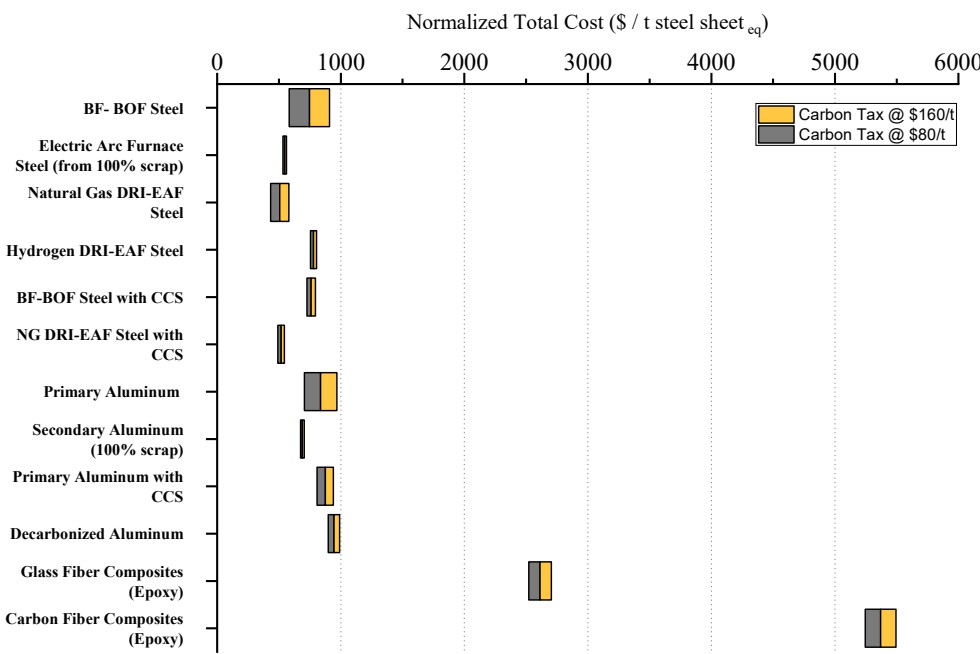

**Figure 11.** Sensitivity analysis of carbon tax on total manufacturing costs (USD/t steel sheet $_{eq}$) (renewables-based grid). Total cost per tonne is defined as the total manufacturing cost of the material plus applied carbon taxes. The costs are normalized according to the equivalent mass method for a sheet product as described in Methods. The midline of each bar depicts the base case of USD 80/t $CO_2$ tax applied on carbon emissions. The left boundary of the gray bar depicts the manufacturing cost with no carbon tax applied. The right boundary of the yellow bar depicts the application of a carbon tax equal to USD 160/t $CO_2$. The LCA assumes a renewables-based electricity grid with a GWP of 0.038 kg $CO_{2\,eq}$/kWh.

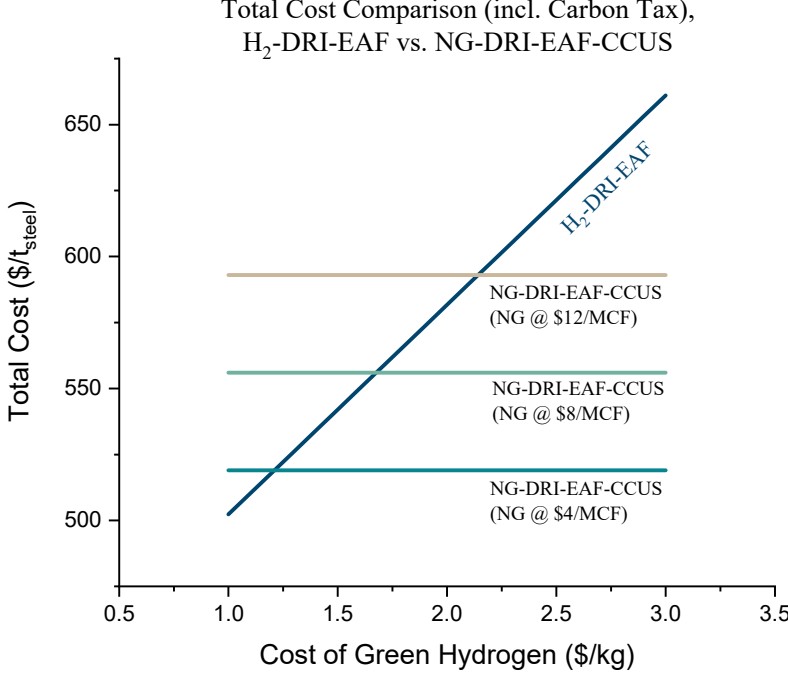

**Figure 12.** Sensitivity analysis of hydrogen-based and natural gas-based DRI technologies to feedstock cost.

## 4. Discussion

The key findings from the life cycle analysis and cost analysis are the following:

The GWP results of the current study for steelmaking and aluminum production are in good agreement with previous studies.

Table 12 compares the GWP calculated and researched in the literature review. The GWP obtained from the literature review has a final product as crude material and for this research, the final product assumed is sheet metal. For steel, the GWP from reheating furnace and sheet rolling is 0.14 t $CO_{2\,eq}$/t steel and for aluminum it is 0.31 t $CO_{2\,eq}$/t Al. The variation in GWP from primary aluminum is seen in the literature due to the variance in the GWP of the electricity grid utilized. The disparities in GWP of glass fiber and carbon fiber production from the literature are because the $CO_2$ emissions from production of these materials depends on production scale and use of different technologies.

**Table 12.** Global Warming Potential of studied materials and processes.

| Process/Material | GWP of Current Studies (t $CO_{2\,eq}$/t Finished Sheet) | | GWP from Literature (t $CO_{2\,eq}$/t Material) | References |
| --- | --- | --- | --- | --- |
| | Traditional Grid | Renewable Grid | | |
| BF- BOF Steel | 2.17 | 2.03 | 1.7–2.5 | [5–7] |
| Electric Arc Furnace Steel (100% scrap) | 0.40 | 0.18 | 0.3–0.47 | [8–12] |
| Natural Gas DRI-EAF Steel | 1.25 | 0.92 | 0.82–1.16 | [13–15] |
| Hydrogen DRI-EAF Steel | 1.64 | 0.31 | 1.1 | [20,21] |
| BF-BOF Steel with CCS | 0.70 | 0.43 | | |
| NG-DRI-EAF Steel with CCS | 0.68 | 0.33 | | |
| Primary Aluminum | 8.22 | 3.28 | 8.2–11.2 | [22–24] |
| Secondary Aluminum (100% scrap) | 0.45 | 0.37 | 0.2–0.85 | [22–24] |
| Primary Aluminum with CCS | 6.84 | 1.65 | | |
| Decarbonized Aluminum | 8.96 | 1.16 | | |
| Glass-Fiber-Reinforced Composite | 1.52 | 0.96 | 1.8–4.6 | [30] |
| Carbon-Fiber-Reinforced Composite | 21.51 | 10.43 | 28.83–42.32 | [28] |
| Resin | 4.7 | 4.7 | 4.9 | [30] |

Under the current US grid mix (considering weight reduction factor) the following conclusions are made:

- The lowest nGWPs belong to secondary aluminum production and the 100% scrap-based EAF route. However, there are two important aspects that make 100% scrap-based EAF scenario only a virtual option for automotive application. First, availability of scrap varies greatly between different geographical locations; the price of high quality (low impurity) scrap also varies. Second, the quality of the steel is tremendously dependent on the scrap elements content. The main problem is caused by the presence of copper in scrap, which is almost impossible to remove. The level of impurities present in the final steel product can vary depending on the steel grade being produced, allowing for copper impurities in the range of 0.04–0.10 wt.% for steel sheet, while for rebar Cu content can be as high as 0.8 wt.% [99]. Current scrap quality achievable almost inevitably requires a high portion of pig iron or DRI for the dilution to obtain automotive sheet steel grade. To achieve the desired quality of the scrap, more efficient sorting is required, which would most likely add cost. Furthermore, the scrap supply is unlikely to meet the worldwide demand for steel to make a significant contribution for worldwide decarbonization.
- In the case of natural gas DRI-EAF with CCS, most of the $CO_2$ emissions from NG can be eliminated by CCS, while the following EAF emissions will be mainly a function of the electricity grid mix. Under the assumption of low-$CO_2$ electricity generation and electrical heating in upstream mining and downstream hot rolling mill, the obtained steel can be produced almost $CO_2$-free. Surprisingly, the $H_2$-DRI-EAF production route has substantially higher GHG emissions compared to NG-DRI when using the current US grid mix. Complete decarbonization of the electricity production

could improve the balance further. It should be mentioned that wide adoption of DRI technologies is not backed with the prospective sufficient supply of necessary DRI-grade iron ore pellets [100].

- BF-BOF with CCS in the present study has only slightly higher emissions than NG-DRI-EAF with the CCS. Such an outcome is obtained due to the assumption that CCS has 90% efficiency, and it is applied to the sintering, blast furnace, and power plant exhaust gases balance of $CO_2$. Indeed, it is technically possible to capture emissions from BF and BOF. However, some parts of the $CO_2$ will be hard to capture (e.g., iron ore mining, ore transportation, hot rolling of the metal sheet).

- Traditional BF-BOF steelmaking routes show relatively high GHG emissions—about 2 t $CO_{2\,eq}$/t steel. However, steel remains competitive compared with alternative materials, such as primary aluminum as well as glass and carbon fibers. Even CCS for primary aluminum will not outcompete traditional steelmaking. Only complete decarbonization of Al production helps it to compete with the current state of steelmaking. However, decarbonization of Al production does not achieve the GHG potential of any decarbonized version of steelmaking. The electrolysis process, also known as the Hall–Héroult process, is the most carbon-intensive during Al production. The process requires a significant amount of electricity, and the electricity used is often generated from fossil fuels, which leads to the emission of carbon dioxide. Recycling of Al can compete with the best steelmaking options in terms of nGWP. When compared to producing aluminum from raw materials, recycling aluminum saves at least 90% of the energy needed for its production. The aluminum recycling rate in the United States is 49.8%, while in Europe it is 76.3% [101]. Increasing recycling incentives could help increase reuse of aluminum, reduce the demand for virgin aluminum production, and therefore, decrease $CO_2$ emissions. However, the same issue arises as for steel recycling, with the dilution requirements with virgin metal for achieving necessary quality.

- $CO_2$ capture and subsequent geological storage is expected to be limited to specific regions with appropriate geological structures. Furthermore, transport of captured $CO_2$ requires significant investment in pipeline and booster pump infrastructure. This transport and sequestration infrastructure currently exists in Texas, USA, so it is insightful to compare hypothetical scenarios involving CCUS there. For illustration's sake, assume one tonne of steel is produced via the hydrogen direct reduction process close to the customer (no transportation GWP or costs). Then consider one tonne of steel produced via natural gas direct reduction, EAF, and CCS in Texas. The latter steel is assumed to be transported by rail freight approximately 2500 km—the average distance from Austin, TX to each of the capitals of the 48 contiguous states. Assume rail transport GWP is 0.015 kg $CO_{2\,eq}$/t per km (primarily diesel fuel combustion) [0.048 lbs $CO_{2\,eq}$/ton-mile, and its cost is USD 0.05/ton-mile [76,102]. The rail transport of the NG-CCS steel from Texas to an "average customer" would add 37 kg $CO_{2\,eq}$/t steel and USD 82/t steel. Neither of these items is significant enough to change our current rank order results showing that NG-DRI-EAF with CCS has lower manufacturing cost and lower total cost (when including an USD 80/t carbon tax) versus hydrogen DRI-EAF per tonne of steel produced. The GWP for NG-DRI-EAF with CCS and rail transport is lower than local $H_2$-DRI-EAF when assuming a traditional electricity grid and 5% higher when assuming the renewables-based grid. This indicates that a regional/national CCS hub with available natural gas could support a steel manufacturing "cluster" which outperforms environmentally and economically against geographically distributed hydrogen DRI siting.

- In the case of glass fiber reinforced composite production, the most carbon-intensive step is resin production. This process involves heating the raw materials followed by melting and refining, requiring a significant amount of energy and resulting in the release of $CO_2$ through the decomposition of the batch materials.

– Due to high baking temperatures (above 1000 °C) and use of natural gas as an energy source, resin production during CFRC manufacturing is considered the most carbon-intensive. About 10.96 kg is emitted to produce 1 kg of carbon-fiber-reinforced composite. If carbon capture technologies would be applied to the resin production, the balance of $CO_2$ might reduce. However, CFRC remains not competitive with steel neither in terms of nGWP nor in terms of normalized cost.

– The solution of the short-term strategy to decarbonize steelmaking processes is highly depended on the local conditions, such as energy grid mix, existing metallurgical infrastructure, potential for CCS, and scrap availability. Government policies and regulations (e.g., carbon pricing, taxes or subsidies for clean energy, and emissions standards) can encourage or mandate the use of low-carbon steelmaking processes. Based on LCA, to radically reduce $CO_2$ emissions from steel it is highly recommended to apply carbon capture to BF-BOF process or use the natural gas DRI-EAF route with carbon capture.

Under the prospective renewable energy grid mix (considering weight reduction factor), the following conclusions are made:

– The best two options, from GHG emission point of view, are steel from recycling (EAF) and recycled aluminum, with an nGWP of a little less than 0.2 kg/t steel equivalent. However, both these options are not the most cost-effective and remain hypothetical for automotive sheet application due to the quality limitations discussed previously and limitations in supply to meet the growing demand.

– NG-DRI-EAF with CCS appears to be the most optimal solution, with 0.33 kg $CO_{2\,eq}$/kg steel and lowest overall cost. In this scenario, the high efficiency of the CCS is removing the lions' share of the $CO_2$ from the DRI production, while decarbonized electricity generation removes the environmental burden of electric arc furnace melting of the DRI and production of steel. The remaining $CO_2$ emissions are the efficiency limit of CCS (0.07 kg $CO_{2\,eq}$/kg steel), $CO_2$ embedded in carbon capture unit (0.04 kg $CO_{2\,eq}$/kg steel), the $CO_2$ of renewable grid mix used for MIDREX shaft reactor operations (0.04 kg $CO_{2\,eq}$/kg steel), EAF (0.06 kg $CO_{2\,eq}$/kg steel) and the upstream emissions (mining and transportation), and those of the hot rolling mill with gas-powered reheating, i.e., 0.07 kg $CO_{2\,eq}$/kg steel.

– The second best cost-performing GWP scenario is $H_2$-DRI-EAF with 0.31 kg $CO_{2\,eq}$/kg steel. Decarbonization of electricity generation contributes substantially to both iron ore reduction by eliminating majority of carbon footprint of electrolysis and to EAF processing of DRI. The remaining balance of $CO_2$ emissions is attributed to electrolysis (0.08 kg $CO_{2\,eq}$/kg steel), EAF (0.06 kg $CO_{2\,eq}$/kg steel), reheating furnace for hot rolling and pelletizing (both 0.06 kg $CO_2$/kg steel), and shaft reactor operations, mining, and transportation (0.01 kg $CO_{2\,eq}$/kg steel for each). Other steelmaking options are at least twice as carbon-intensive as NG-DRI-EAF with CCUS and $H_2$-DRI-EAF routes. The assumed cost of hydrogen is one of the most important cost drivers in the hydrogen DRI scenario. As discussed previously, there is great uncertainty around the cost of scaling up the necessary electrical generation and electrolyzer capacity. We have selected a middle value of USD 4.5/kg for total landed cost of hydrogen (Figure 12). This is far above the United States Department of Energy "Hydrogen Shot" target of USD 1/kg [103]. Achieving that target would dramatically reduce the cost of the $H_2$-DRI-EAF process and would make this production route by far the most optimal. Aside from cost reduction potential, production of hydrogen is more flexible towards geographic location compared to CCUS projects needed for NG-DRI. These two aspects imply that $H_2$-DRI-EAF has a better long-term potential over NG-DRI-EAF with CCUS and should be prioritized as an investment strategy.

– Recycled aluminum showed emissions of 0.19 kg $CO_{2\,eq}$/kg steel $_{eq}$. Most of the remaining carbon footprint is coming from the scrap collection and transportation, and from the smelting of the scrap and rolling the Al sheet. Maximum decarbonized primary aluminum production is still lagging behind the recycling route with

0.58 kg of $CO_2$ eq/kg steel eq. Even primary aluminum production of Al sheet with CCS, due to the renewable grid, substantially improves its carbon footprint from 3.42 kg to 0.83 kg of $CO_2$ eq/kg steel eq. However, it remains among the heaviest from environmental footprint.

- Both glass fiber and carbon fiber composite materials production require implementation of carbon capture solutions along the production chain to be competitive in terms of nGWP potential with the decarbonized steelmaking and with decarbonized aluminum. Even without CCUS implementation, these materials are not cost-competitive with steel and aluminum.

## 5. Conclusions

The restrictions on greenhouse gas emissions play a crucial role in the marketplace, which can shift according to individual manufacturers' abilities to reduce carbon dioxide emissions as measured by Life Cycle Analysis (LCA). The current study has a comparative cost analysis for automotive sheet as a target product. The potential of different pathways for the decarbonization of steel sheet and alternative materials, i.e., aluminum, carbon-fiber-reinforced composite and glass-fiber-reinforced composite was studied.

An average current US electricity grid mix and conventional BF-BOF process for producing automotive sheet was used as a baseline with a GWP potential of 2.174 $CO_2$ eq/kg hot-rolled sheet. The GWPs, normalized for automotive sheet for different decarbonization strategies were found to be the following:

- When comparing various steel decarbonization methods, in order of highest to lowest impact on GWP reduction, 100% scrap-based production through EAF resulted in an 81.5% reduction, followed by natural gas DRI-EAF with carbon capture with a 68.8% reduction, and BF-BOF with carbon capture (CCS) with 68%. Hydrogen-based DRI with EAF offers only a 24.6% reduction owing to the required electric power derived from current grids. Among them, natural gas DRI-EAF (USD 533/t) and natural gas DRI-EAF with CCS (USD 546/t) as well as 100% scrap-based EAF (USD 564.5/t) are the cheapest.
- Using 100% recycling in production of Al is the most environmentally friendly option for sheet decarbonization with the potential to cut baseline GWP by 90% by achieving nGWP of 0.22 $CO_2$ eq/kg hot-rolled sheet. Secondary Al (USD 668.25/t) is the cheapest, while primary Al with CCS (USD 1082.55/t) and decarbonized primary Al (USD 1257.15) are too expensive to compare to steel equivalent.
- Glass-fiber-reinforced composite with a nGWP of 1.39 $CO_2$ eq/kg hot-rolled sheet instead of conventional steel has a potential to reduce GWP by 36%. At the same time, GFRC and CFRC are not cost-competitive with their production costs of USD 2633.88/t and USD 5677/t, respectively.
- Neither primary aluminum nor carbon-fiber-reinforced composites offer decarbonization potential compared to the current baseline of BF-BOF without carbon capture.

When considering a renewable electricity grid of 50% solar photovoltaic and 50% wind, and BF-BOF as a baseline with its emissions of 2.03 $CO_2$ eq/kg hot-rolled sheet (corresponding to a 6.6% GWP reduction to current conditions), the GWP for different steel decarbonization strategies were found to be the following:

- When comparing various steel decarbonization methods, in order of highest to lowest impact on GWP reduction, 100% scrap-based EAF steel remained the highest with a 91% reduction, followed by hydrogen-based DRI-EAF (84.7% reduction), and natural gas DRI-EAF with CCUS (83.7% reduction) and BF-BOF with CCUS (79% reduction). Natural gas DRI-EAF (USD 506.8/t) and natural gas DRI-EAF with CCS (USD 517.5/t) as well as 100% scrap-based EAF (USD 546.7/t) remain the cheapest;
- Using 100% recycling of Al is on par with 100% steel recycling, resulting in a 91% nGWP reduction;
- Primary aluminum becomes more favorable when a renewable grid is used. Without any decarbonization technologies, it has an nGWP of 1.64 $CO_2$ eq/kg hot-rolled sheet

eq, which is a 19% decrease from the baseline BF-BOF route. Application of CCS lowers $CO_2$ emissions to 0.83 $CO_{2\,eq}$/kg hot-rolled sheet eq (59% decrease). Complete decarbonization of Al with 0.58 $CO_{2\,eq}$/kg hot-rolled sheet eq brings the emissions down 71.4%. Although secondary Al (USD 665.1/t) is still the cheapest Al production route, primary Al with CCS (USD 875.1/t) and decarbonized primary Al (USD 945.4) will also significantly drop in price with the renewable grid mix.

- Carbon-fiber-reinforced composite (CFRC) offers about a 24% decrease (1.55 $CO_{2\,eq}$/kg hot-rolled sheet eq);
- Glass-fiber-reinforced composite (GFRC) offers about a 43% reduction of nGWP with 1.15 $CO_{2\,eq}$/kg hot-rolled sheet eq;
- Costs for future decarbonized production pathways of GFRC and CCRC are USD 2614.04/t and USD 5369.35/t, respectively, which are prohibitive for mass production.

The results indicate that, when applying technologies available today, decarbonized steel will remain competitive, at least in the context of automotive sheet selection compared to aluminum and composites.

Under the assumption of hydrogen cost decreasing from 4.5 USD/kg into the region of 1.5–2.5 USD/kg $H_2$-DRI-EAF steel becomes more cost-competitive than NG-DRI with CCUS.

**Supplementary Materials:** The supporting information can be downloaded at: https://www.mdpi.com/article/10.3390/en16196904/s1.

**Author Contributions:** Conceptualization, V.V.R., T.S., R.S., T.J., Y.K., S.S., S.B., B.B. and D.B.; methodology, S.S., T.S., V.V.R. and R.S.; software, V.V.R., R.S., T.S. and Y.K.; validation, S.S., S.B., B.B. and D.B.; formal analysis, V.V.R., R.S., T.S. and Y.K.; investigation, V.V.R., R.S., T.S., Y.K. and T.J.; writing—original draft preparation, V.V.R., T.S., R.S., T.J. and Y.K.; writing—review and editing, S.S., S.B., B.B. and D.B.; supervision, S.S. and D.B. All authors have read and agreed to the published version of the manuscript.

**Funding:** This work was partially funded by the Tata Steel Ltd.

**Data Availability Statement:** The data presented in this study are available in the supplementary materials.

**Acknowledgments:** The authors wish to thank Ankit Singhania for his contribution to this research effort. The authors wish to thank Jenny Sumner at the National Renewable Energy Laboratory (NREL) for guidance on Renewable Energy Credits.

**Conflicts of Interest:** The authors declare no conflict of interest.

## Abbreviations

| | |
|---|---|
| AEL | Alkaline Electrolysis |
| BF | Blast Furnace |
| BMI | Bismaleimide |
| BOF | Basic Oxygen Furnace |
| CCS | Carbon Capture and Storage |
| CCUS | Carbon Capture, Usage, and Storage |
| CFRC | Carbon-Fiber-Reinforced Composite |
| $CO_2$ | Carbon Dioxide |
| CPI-U | Consumer Price Index for all Urban Consumers |
| DRI | Direct Reduced Iron |
| E | Elastic modulus (GPa) |
| EAF | Electric Arc Furnace |
| E-glass | Electrical Glass |
| $_{eq}$/t | Equivalent/tonne |
| (g) | The substance is in a gaseous state |
| GFRC | Glass-Fiber-Reinforced Composite |
| GWP | Global Warming Potential |

| H$_2$ | Hydrogen gas |
| IEA | International Energy Agency |
| K | Stiffness (N/m) |
| kWh/t | Kilowatt-hours/tonne |
| LCA | Life Cycle Analysis |
| MEA | Mono-Ethanolamine |
| Mfg. Cost | Manufacturing Cost |
| Mtpa | Million tonnes per annum |
| NG | Natural Gas |
| nGWP | normalized Global Warming Potential |
| O&G | Oil and Gas |
| PAN | Polyacrylonitrile |
| PEMEL | Polymer Electrolyte Membrane Electrolysis |
| (s) | The substance is in a solid state |
| SMC | Sheet Molding Compound |
| SOEL | Solid Oxide Electrolysis |
| t | Sheet thickness (cm) |
| t [unit] | Metric ton; tonne |
| TEA | Techno-Economic Analysis |

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
