# Peer review of "Steel, Aluminum, and FRP-Composites: The Race to Zero Carbon Emissions"

_energies, doi:10.3390/en16196904_

Round 1

Reviewer 1 Report

For a paper to be published in 2023, the data was taken from 2019, or even 2013. Are those numbers still valid now?

For example, in Table 2, the 2021 data is published or forecast? If they were forecast by earlier literatures, what are the actual numbers?

Similarly, in Table 8, the latest date is from 8 years ago. Is carbon capture technique stalled in this decade?

In general, some validation is necessary to prove this paper is not outdated.

There are some manuscript errors:

In line 285, “Error! Reference source not found”.

In line 381, “Error! Not a valid bookmark self-reference.”

In line 633, “Error! Reference source not found.”

No Comment

Author Response

Dear reviewer,

Thank you for your evaluation of our work!

All important corrections were made.
(Please, ignore the attached file as the reviewers' order has been changed and I can't delete the file that I uploaded).

To your comments:

For a paper to be published in 2023, the data was taken from 2019, or even 2013. Are those numbers still valid now?

The most recent prices are added as well as an explanation why the year 2019 was used as the basis for analysis. Please, see lines 414-425. 

For example, in Table 2, the 2021 data is published or forecast? If they were forecast by earlier literatures, what are the actual numbers?

Table 2 is presented for costs comparisons. In our study, year 2019 was chosen as the basis for analysis as it is representative of manufacturing costs in the years before the global COVID-19 pandemic and Ukraine conflict. Although labor, feedstock, and energy prices always change, we assert that the 2019 prices cited in this work are representative of long-term, stable prices.

Between early-2019 and mid-2022, many input (e.g. ore, energy) and finished steel prices climbed dramatically due to supply chain volatility. Current commodity values seem to be returning to values somewhat similar to our cost assumptions. For example, we assume iron ore prices of $107/t, the average for 2019. Iron ore prices peaked at $220/t in May 2021, but have returned to $112/t as of June 2023. Likewise, we assume natural gas prices of $3.84/million cubic feet (MCF, USA), which was the 2019 average industrial price. June 2023 US Industrial prices are $3.64/MCF after reaching peaks of $9.95/MCF in September 2022.

Similarly, in Table 8, the latest date is from 8 years ago. Is carbon capture technique stalled in this decade?

In table 8, there is also the data from 2020.

In general, some validation is necessary to prove this paper is not outdated.

There are some manuscript errors:

All references were checked and no errors were found. Please, let us know if you still have errors.

In line 285, “Error! Reference source not found”.

In line 381, “Error! Not a valid bookmark self-reference.”

In line 633, “Error! Reference source not found.”

Reviewer 2 Report

Based on life cycle analysis, the potential of different decarbonization pathways for steel plate and alternative materials such as aluminum, carbon fiber reinforced composite and glass fiber reinforced composite was investigated. For a current electricity grid mix in the US (with 61 % fossil fuels, 19 % nuclear, 20 % renewables), the lowest nGWP was found to be secondary aluminum and 100 % recycled scrap melting of steel. This is followed by natural gas DRI-EAF route with carbon capture and BF-BOF route with carbon capture. From the cost point of view, the current cheapest decarbonized production route is natural gas DRI-EAF with CCS. For a renewable electricity grid (50 % solar photovoltaic and 50 % wind), the lowest GWP was found to be 100 % recycled scrap melting of steel and secondary aluminum. It has certain research and extension value. The article also has the following issues that need to be modified or improved.

1. Line 179 of the article only says that the carbon footprint of EAF steelmaking is low, but there is no data support, it is suggested to add the data of EAF steelmaking to compare with the BF converter process.

2. Please correct the error on line 285, 332, 381, etc.

3.The text in Figure 5 is not clear, please modify it.

4. The suggestion in line 401 of the article can provide a price list in recent years, which is more convincing.

5. The article does not have a literal description of Table 12, please update.

Author Response

Dear reviewer,

Thank you for your evaluation of our work! Please, find the replies to your comments in the attached file. All requested corrections are made.

Reviewer 3 Report

Dear Authors,

Your proposition is attractive. I have some concerns, but after You resolve them I think your paper can be published, thus I recommend major revisions. 

My concerns are split into minor and major:

MINOR

* Figures 1 and 2 are chaotic, not very aesthetic and thus not very readable. Please redraw it - maybe reorganize, align, use similar width of rectangles, check if things are not overlapping or other means... // Also applicable for next figures like 3

* traditional electricity values in Table 1 are cited from some sources, but what with renewable ones? What is the source for them?

* On page 7/8 there is not named table (aligned to left?) - it seems like an editing error

* page 12 - lacking reference - probably an error connected to some reference manager - please repair it

* Table 5 is present twice in the text (another in-word reference mistake?) 

* Figure 5 - has bad quality - maybe change the font, and save it in better quality

* page 15 - another reference error

* Table 7 is not mentioned in the text

MAJOR

* I am concerned with subsection 2.9 - Normalization using Material Properties. Composite materials are a very important group of materials nowadays, however, you propose to compare steel to them and other materials like aluminium only using bending stiffness. It is of course important factor but not the only one. For material properties, we have to consider aspects like fatigue behaviour, fracture properties, presence of plasticity etc. Also, composites in contrary to steel are not isoptropic materials, which has to be taken into account. This section has to be much extended and not only limited to bending stiffness.

* Figures 1 and 2 are chaotic, not very aesthetic and thus not very readable. Please redraw it - maybe reorganize, align, use similar width of rectangles, check if things are not overlapping or other means... // Also applicable for next figures like 3

* It may be worth mentioning combined materials that can be used in the automotive industry - fibre metal laminates which consist of both metal and frp-composite layers. They can be based on steel or aluminium. An important factor is that for the automotive industry they should be based on thermoplastic resin - for example, you can find examples of work on them here https://doi.org/10.1016/j.compstruct.2022.115810, but you should be able to find also some analysis about FMLs in the specific area of your paper (CO2 cost etc). 

* Conclusions could be easier to follow - maybe you can consider some graphical summary?

The paper is written with good quality. Some minor mistakes in editing are present, so just re-read the paper and improve them.

Author Response

(The authors gave the same response as above.)

Reviewer 4 Report

The paper compares the carbon dioxide emissions for various processes to discuss the potential of decarbonised materials used in automotive industry. Overall, in my opinion the paper is comprehensive and is worth publishing. However, there are some changes that would help to improve it.

Introduction:

Paragraph lines 52-60, should be moved to the end of the introduction section. The novelty of this paper should be clearly highlighted.

The introduction is quite difficult to read. Presenting this data in the table will help the reader.

Table 5 Did the authors include the glass-fibre production (Figure 4) in CO2 emission for production of 1 kg of GFRC?

Table 7 – typo error reinforcement

Line 484 could you please include the reference/datasheet which from the material properties where taken.

Line 492 typos

Line 493 The bending stiffness should not be considered proportional to cube of thickness for every auto component. That means, the weight saving is not the same in every case. Even, in some cases, there may be weight gain as well. – This statement is not clear please revise.

Line 510 typo

4. Discussion

Line 647 is missing headings and captions.

Line 751 glass fibre composite production

Throughout the article:

Please make sure all abbreviations are spelled out correctly and consistently. They are typos in CO2 spelling.

Table of abbreviations would be useful.

Please check all the references and in-text citations.

Author Response

(The authors gave the same response as above.)

Round 2

Reviewer 1 Report

Authors have answered my concerns

No comment